# A new assessment of global and regional budgets, fluxes and lifetimes of atmospheric reactive N and S gases and aerosols

Yao Ge[1,2], Massimo Vieno[2], David S. Stevenson[3], Peter Wind[4], Mathew R. Heal[1]

[1] School of Chemistry, University of Edinburgh, Joseph Black Building, David Brewster Road, Edinburgh, EH9 3FJ, UK

[2] UK Centre for Ecology & Hydrology, Bush Estate, Penicuik, Midlothian, EH26 0QB, UK

[3] School of GeoSciences, University of Edinburgh, Crew Building, Alexander Crum Brown Road, Edinburgh, EH9 3FF, UK

[4] The Norwegian Meteorological Institute, Henrik Mohns Plass 1, 0313, Oslo, Norway

*Correspondence to*: Yao Ge (Y.Ge-7@sms.ed.ac.uk), Mathew R. Heal (M.Heal@ed.ac.uk)

**Abstract.** We used the EMEP MSC-W model version 4.34 coupled with WRF model version 4.2.2 meteorology to undertake a present-day (2015) global and regional quantification of the concentrations, deposition, budgets, and lifetimes of atmospheric reactive N ($N_r$) and S ($S_r$) species. These are quantities that cannot be derived from measurements alone. In areas with high levels of reduced $N_r$ (RDN = $NH_3$ + $NH_4^+$), oxidised $N_r$ (OXN = $NO_x$ + $HNO_3$ + HONO + $N_2O_5$ + $NO_3^-$ + Other OXN species), and oxidised $S_r$ (OXS = $SO_2$ + $SO_4^{2-}$), RDN is predominantly in the form of $NH_3$ ($NH_4^+$ typically <20%), OXN has majority gaseous species composition, and OXS predominantly comprises $SO_4^{2-}$ except near major $SO_2$ sources. Most continental regions are now 'ammonia rich', and more so than previously, which indicates that whilst reducing $NH_3$ emissions will decrease RDN concentration it will have small effect on mitigating SIA. South Asia is the most ammonia-rich region. Coastal areas around East Asia, northern Europe, and north-eastern United States are 'nitrate rich' where $NH_4NO_3$ formation is limited by $NH_3$. These locations experience transport of OXN from the adjacent continent and/or direct shipping emissions of $NO_x$ but $NH_3$ concentrations are lower. The least populated continental areas and most marine areas are 'sulfate rich.' Deposition of OXN (57.9 TgN yr$^{-1}$, 51%) and RDN (55.5 TgN yr$^{-1}$, 49%) contribute almost equally to total nitrogen deposition. OXS deposition is 50.5 TgS yr$^{-1}$. Globally, wet and dry deposition contribute similarly to RDN deposition; for OXN and OXS, wet deposition contributes slightly more. Dry deposition of $NH_3$ is the largest contributor to RDN deposition in most regions except for Rest of Asia and marine areas where $NH_3$ emissions are small and RDN deposition is mainly determined by transport and rainout of $NH_4^+$ (rather than rainout of gaseous $NH_3$). Reductions in $NH_3$ would thus efficiently reduce the deposition of RDN in most continental regions. The two largest contributors to OXN deposition in all regions are $HNO_3$ and coarse $NO_3^-$ (via both wet and dry deposition). The deposition of fine $NO_3^-$ is only important over East Asia. The tropospheric burden of RDN is 0.75 TgN of which $NH_3$ and $NH_4^+$ comprise 32% (0.24 TgN; lifetime = 1.6 days) and 68% (0.51 TgN; lifetime = 8.9 days), respectively. The lifetime of RDN (4.9-5.2 days) is shorter than that of OXN (7.6-7.7 days), consistent with a total OXN burden (1.20 TgN) almost double that of RDN. The tropospheric burden of OXS is 0.78 TgS with a lifetime of 5.6-5.9 days. Total nitrate burden is 0.58 TgN with fine $NO_3^-$ only constituting 10% of this total, although fine $NO_3^-$ dominates in eastern China, Europe, and eastern North America. It is important to account for contributions of coarse nitrate to global nitrate budgets. Lifetimes of RDN, OXN, and OXS species vary by a factor of 4 across different continental regions. In East Asia, lifetimes for RDN (2.9-3.0 days), OXN (3.9-4.5 days), and OXS (3.4-3.7 days) are short, whereas lifetimes in Rest of Asia and Africa are about twice as long. South Asia is the largest net exporter of RDN (2.21 TgN yr$^{-1}$, 29% of its annual emission), followed by Euro_Medi region. Despite having the largest RDN emissions and deposition, East Asia has only small net export and is therefore largely responsible for its own RDN pollution. Africa is the largest net exporter of OXN (1.92 TgN yr$^{-1}$, 22%), followed by Euro_Medi (1.61 TgN yr$^{-1}$, 26%). Considerable marine anthropogenic $N_r$ and $S_r$ pollution is revealed by the large net import of RDN, OXN and OXS to these areas. Our work demonstrates the substantial regional variation in $N_r$ and $S_r$ budgets and the need for modelling to simulate the chemical and meteorological linkages underpinning atmospheric responses to precursor emissions.

# 1 Introduction

Reactive N ($N_r$) and S ($S_r$) gases and particle species are critical determinants of air quality. The principal precursor gas emissions of NO and $NO_2$ (collectively $NO_x$), $NH_3$ and $SO_2$ are emitted from a wide range of human activities (Crippa et al., 2018; Hoesly et al., 2018). As the dominant alkaline gas in the atmosphere, $NH_3$ exerts significant control on the acidity of deposition and on the formation of the secondary inorganic aerosol (SIA) components of fine particles ($PM_{2.5}$) (Sutton et al., 2020). The latter derive from the reactions between $NH_3$ and the sulfuric acid ($H_2SO_4$) and nitric acid ($HNO_3$) that are oxidation products of $SO_2$ and $NO_x$ (Xue et al., 2011; Behera et al., 2013a; Nenes et al., 2020). These particles have significant impacts on human health (Lelieveld et al., 2015; Cohen et al., 2017; Burnett et al., 2018) and on Earth's radiation budget (Bellouin et al., 2011; IPCC, 2021). Sulfur dioxide and sulfate aerosols (collectively referred to as oxidised S and abbreviated as OXS) also have an impact on cloud formation and the acidity of precipitation (Myhre et al., 2017; Aas et al., 2019; Thornhill et al., 2021; McHale et al., 2021).

In addition, oxidised N gases and aerosols such as $NO_x$, $HNO_3$, $NO_3^-$, etc. (collectively abbreviated as OXN), and reduced N gases and aerosol ($NH_3$ and $NH_4^+$, collectively abbreviated as RDN), act as powerful plant and microorganism nutrients when deposited to aquatic and terrestrial ecosystems. Excessive anthropogenic input of N can lead to exceedance of N critical loads, which in turn can lead to eutrophication and loss of ecosystem productivity and biodiversity (Erisman et al., 2005; Fowler et al., 2013; Sun et al., 2017). The increase in food production to support a burgeoning human population is increasing agricultural emissions of $NH_3$ globally, in contrast to the globally declining emissions of $SO_2$ and $NO_x$ (Fowler et al., 2020). A further factor contributing to the different trend in $NH_3$ emissions is the lack of regulations in most countries specifically targeted at reducing $NH_3$ emissions. The relative proportions of the $N_r$ and $S_r$ gas and aerosol species in the atmospheric burdens and wet and dry deposition of RDN, OXN and OXS vary substantially around the world because of the different magnitudes of precursor emissions and local meteorology. Knowledge of how these burdens and depositions vary regionally is important for understanding local impacts and mitigation actions.

This information can be derived from atmospheric chemistry transport models. An advantage of these models is that they can simulate concentrations and depositions of air pollutants at greater spatial and temporal resolution and coverage than can be achieved through ambient measurements. Furthermore, since these models simulate the underlying physical-chemical processes, they allow insight into the chemical and meteorological linkages between emissions, concentrations and deposition of pollutants that measurements cannot reveal: for example, horizontal and vertical pollutant transport, and the proportion of reduced nitrogen in precipitation that derives from rainout of gaseous $NH_3$ or particulate $NH_4^+$. A number of global-scale modelling studies of atmospheric $N_r$ and $S_r$ have been undertaken previously. Lamarque et al. (2013) presented a multi-model ensemble of N and S deposition from 1850 to 2100, derived from the Atmospheric Chemistry and Climate Model Intercomparison Project (ACCMIP). The ACCMIP results for 'present day' (year 2000 ACCMIP time slice) showed a multi-model mean deposition of 110 $TgN\ yr^{-1}$, and 83 $TgS\ yr^{-1}$. Hauglustaine et al. (2014) reported that 'present-day' (2000) fine nitrate and ammonium particles make important contributions to aerosol burdens and direct radiative forcing. Bian et al. (2017) conducted an evaluation of global $NO_3^-$ aerosol in 2008 from 9 models involved in the Aerosol Comparisons between Observations and Models (AeroCom) phase III and reported a large diversity in global tropospheric burdens of $HNO_3$, $NO_3^-$, $NH_3$ and $NH_4^+$ between the models. Tan et al. (2018) calculated the global S and N deposition in 2010 utilizing results of 11 models from the second phase of the Task Force on Hemispheric Transport of Air Pollution (HTAP II). The results showed that global S and N depositions in 2010 were 84 $TgS\ yr^{-1}$ and 123 $TgN\ yr^{-1}$, respectively. Comparisons of their results with those from HTAP I indicated that global distributions of S and N deposition had changed considerably between 2001 and 2010. Globally there was a reduction in S deposition of 2 TgS (3%) during this period, with significant reductions in S deposition in Europe, North America, and Russia. In contrast, global N deposition increased by 7 TgN (6%), particularly in South Asia, East

Asia, and Southeast Asia. Ackerman et al. (2019) used the GEOS-Chem Chemical Transport Model to estimate global inorganic N deposition from 1984 to 2016 and found an 8% increase from 86.6 to 93.6 TgN yr$^{-1}$.

These previous modelling studies provide an outlook on the distribution of sulfate-ammonium-nitrate aerosols in the early 2000s with predominant focus on S and oxidised N. However, a more up-to-date examination of the global state of $N_r$ and $S_r$ deposition, atmospheric burden, deposition and lifetime is needed, particularly of the reduced $N_r$ species $NH_3$ and $NH_4^+$, which has not previously been comprehensively conducted. In recent years there have been large changes in global and regional $NH_3$, $NO_x$ and $SO_x$ emissions, especially over South Asia, East Asia, Europe, and North America (Crippa et al., 2018; Hoesly et al., 2018). For instance, Kurokawa and Ohara (2020) reported that emissions of $SO_2$ and $NO_x$ in East Asia increased rapidly from 1950 to 2015, reaching their peaks during 2005-2015, yet $NH_3$ emissions continued to increase. On the other hand, $NO_x$ and $SO_2$ emissions in South Asia have continued to increase between 1996 and 2015 (Sadavarte and Venkataraman, 2014; Pandey et al., 2014; Kurokawa and Ohara, 2020). In Europe and the United States there has been a general decreasing trend of N and S emissions is in the last decade (Westervelt et al., 2017; UNECE, 2020; Westervelt et al., 2021).

An updated global modelling of $N_r$ and $S_r$ gases and aerosols with the latest available emission inventory is crucial to understand current differences in gas-aerosol partitioning, wet and dry deposition, tropospheric burden and lifetime of separate RDN, OXN, and OXS species in separate regions, and as a basis for developing mitigation strategies according to regional conditions. In this study, we use the three-dimensional global EMEP MSC-W model 4.34 coupled with WRF 4.2.2 meteorology to calculate the atmospheric concentration, deposition, and tropospheric budget of $N_r$ and $S_r$ gases and aerosols in 2015. Section 2 details the framework of our model simulation including emission, meteorology, and aerosol scheme. The global distribution and typical magnitude of $N_r$ and $S_r$ concentration, regional deposition processes of RDN ($NH_3$, $NH_4^+$), OXN ($NO_x$, $HNO_3$, $HONO$, $N_2O_5$, fine $NO_3^-$, coarse $NO_3^-$, other OXN species) and OXS ($SO_2$, $SO_4^{2-}$), tropospheric budgets and lifetimes of major $N_r$ and $S_r$ species are described in Sect. 3. We summarise the conclusions of our study in Sect. 4.

## 2 Methods

### 2.1 Model description

We used version rv4.34 of the EMEP MSC-W (European Monitoring and Evaluation Programme Meteorological Synthesizing Centre –West) open-source atmospheric chemistry transport model (www.emep.int), which is widely used in scientific research and policy development (Simpson et al., 2012; Bergström et al., 2012; Bergström et al., 2014; Jonson et al., 2017; Karl et al., 2019; McFiggans et al., 2019; Fagerli et al., 2019; Pommier et al., 2020). The model was driven by meteorology from the Weather Research and Forecast model (WRF; www.wrf-model.org; https://github.com/wrf-model/WRF/releases/tag/v4.2.2) version 4.2.2 at 1° grid resolution (Vieno et al., 2010, 2014, 2016; Gu et al., 2021). The WRF simulation in this work assimilates data of the numerical weather prediction model meteorological reanalysis from the US National Center for Environmental Prediction (NCEP)/National Center for Atmospheric Research (NCAR) Global Forecast System (GFS) (Saha et al., 2010). The global EMEP-WRF configurations are detailed in Ge et al. (2021).

A detailed technical description of EMEP MSC-W rv4.0 is documented in Simpson et al. (2012). An overview of model updates from version rv4.0 to rv4.34 is presented in Simpson et al (2020). The model has horizontal resolution of 1° × 1° and contains 21 terrain-following vertical layers from the surface to 100 hPa. The lowest model layer has height ~50 m and surface concentrations of most species were adjusted to correspond to 3 m above the surface as described in Simpson et al. (2012; 2020).

The model aerosol scheme is the Equilibrium Simplified Aerosol Model V4 (EQSAM4clim) (Metzger et al. 2012; 2016; 2018), which efficiently parameterises the aerosol water uptake and full gas-liquid-solid partitioning of mixtures of semi-volatile and non-volatile compounds. In addition to the $SO_4^{2-}$-$NO_3^-$-$NH_4^+$ scheme, the model includes the reactions between

HNO$_3$ and dust and sea salt to produce coarse nitrate. The aerosol scheme is described in more detail in Supplementary Material Sect. S1.1.

Dry deposition of gaseous species is simulated utilizing deposition velocity as described in Simpson et al. (2012; 2020).
Aerosol dry deposition follows similar formulation to other studies (Lamaud, 1994; Gallagher et al., 1997; Nemitz et al., 2004), with modification for nitrogen compounds (Petroff et al., 2008b, a; Simpson et al., 2012). For wet deposition, all generic fine particles (e.g., SO$_4^{2-}$, NH$_4^+$, NO$_3^-$_f, etc.) have the same wet scavenging ratio (in-cloud) and size-dependent collection efficiency of aerosols by raindrops (below-cloud), whilst coarse particles are divided into two groups (coarse sea salt and other coarse particles) with their own sets of parameters (Berge and Jakobsen, 1998; Laakso et al., 2003; Simpson et al., 2012).

**2.2 Emissions**

Anthropogenic emissions were derived from V6 of the ECLIPSE (Evaluating the CLimate and Air Quality ImPacts of Short-livEd Pollutant) emission inventory (https://iiasa.ac.at/web/home/research/researchPrograms/air/ECLIPSEv6.html). This contains global annual emissions for 2015 of SO$_2$, NO$_2$, NH$_3$, CO, CH$_4$, NMVOC (non-methane volatile organic compounds), primary PM$_{2.5}$ and primary coarse particulate matter (PM$_{co}$) (Klimont et al., 2017) at a horizontal resolution of
35 0.5°. All emissions were aggregated to 1° resolution internally in the model. The ECLIPSE emission sector-layers were re-assigned to 11 Selected Nomenclature for sources of Air Pollution (SNAP) sectors (Ge et al., 2021). Monthly emission profiles by country/region and by emission sector based on EDGAR (Emission Database for Global Atmospheric Research, v4.3.2 datasets) time series (Crippa et al., 2020) (https://edgar.jrc.ec.europa.eu/dataset_temp_profile) were then applied to the ECLIPSE annual emissions for each pollutant. The calculation of the time-splitting factors for a given pollutant and
40 country/region is given in Ge et al. (2021). The terminology ECLIPSE$_E$ is used here to refer to these monthly emissions.

Dimethyl sulphide (DMS), lightning NO$_x$, soil NO$_x$ and isoprene emissions were configured as reported in Simpson et al. (2017; 2020) as were the wind-derived emissions of sea salt particles and dust (Tsyro et al., 2011; Simpson et al., 2012). Emissions from forest and vegetation fires, and international shipping were also included.

**2.3 Model performance**

Ge et al. (2021) provide a comprehensive evaluation of surface concentrations and wet deposition of N$_r$ and S$_r$ species from this model configuration against global measurements from 10 monitoring networks. Comparisons of 2010 and 2015 surface concentrations between model and measurement demonstrate that the model captures well the overall spatial and seasonal variations of the major inorganic pollutants NH$_3$, NO$_2$, SO$_2$, HNO$_3$, NH$_4^+$, NO$_3^-$, SO$_4^{2-}$, and their wet deposition in East Asia, Southeast Asia, Europe, and North America. Correlation coefficients between modelled and measured annual mean
concentrations of all species are >0.78 except for HNO$_3$ and SO$_2$, and are 0.78 and 0.63 for annual wet deposition of reduced N and oxidised N, respectively. Given the intrinsic scale mismatch between a local site measurement and a global-scale chemistry model grid, these comparisons are good, and are comparable with model evaluation statistics determined for models of similar resolution (Hauglustaine et al., 2014; Bellouin et al., 2011; Pringle et al., 2010; Xu and Penner, 2012).

Both model and measurement have uncertainty that constrains the extent to which statistical analyses between modelled
and measured data can be utilized to assess a model's performance. Model uncertainty comes from uncertainty in emission and meteorology inputs and from parameterisations of chemical/physical processes, while measurement uncertainty comes from sampling and analytical procedures. Emission inventory compilation is partially based on reported measurement data and partially on expert estimation, which complicates assigning uncertainties to both emission magnitudes and their spatiotemporal profiles. It is therefore not possible to quantify the contribution of emissions uncertainty to model uncertainty, but it will surely
vary regionally and for different species. We have demonstrated in Ge et al. (2021) that modelled concentrations of secondary pollutants are less sensitive to the choice of emission inventory than for primary pollutants since the former are influenced by

multiple emissions and the timescales for their formation act to smooth out the differentials in primary emissions. On the other hand, if it is variation in measurement accuracy that contributes to regional variation in model-measurement agreement, then confidence in model output is maintained. Ge et al. (2021) showed better evaluation statistics with measurements in Southeast Asia, Europe, and North America than in East Asia, which suggests shortcomings in regional measurements and/or emissions in the latter network rather than substantial systematic issues with model parameterisations of chemical/physical processes. Ge et al. (2021) also evaluated model response to changes between 2010 and 2015 ECLIPSE$_E$ emissions and showed that modelled concentration changes of primary and secondary pollutants and of deposition of reduced N, oxidised N, and oxidised S, were consistent with the trends in the corresponding measurements, given realistic uncertainties in emissions and measurements and of meteorological influences between the two years.

Nevertheless, considering the fundamental uncertainties in emission, model and measurement, all numbers reported in this work should be considered as having underlying uncertainty, albeit that the latest available emission inventory and model version were used to minimize as far as possible uncertainties from the first two of these.

## 2.4 Definition of groups of chemical species

The following terminology is used to refer to groups of $N_r$ and $S_r$ species. RDN represents atmospheric reduced N and is the sum of $NH_3$ and $NH_4^+$. OXN represents all oxidised N outputs in the model and consists of $HNO_3$, $NO_x$, $NO_3^-$_f, $NO_3^-$_c and 'Rest_OXN.' The latter comprises $N_2O_5$, HONO, and 'Other_OXN' species, which is mainly composed of organic oxidised N species, of which the predominant component is PAN (peroxyacetylnitrate). Full details are given in Table S1. Total reactive N is the sum of RDN and OXN. OXS represents oxidised S and is the sum of $SO_2$ and total $SO_4^{2-}$.

In this paper, all concentrations, depositions, emissions, and burdens are expressed as mass of the species (e.g., $\mu g\ m^{-3}$, $mg\ m^{-2}$) unless otherwise specified (e.g., $\mu gN\ m^{-3}$, $\mu mol\ m^{-3}$, $TgN\ yr^{-1}$).

## 2.5 Definition of world regions

To compare atmospheric budgets and deposition of $N_r$ and $S_r$ species around the world, we divide the world into the 10 regions illustrated in Fig. 1 (and listed in Table S2). These are based on the 58 reference land and ocean regions defined by the Intergovernmental Panel on Climate Change (IPCC) (Iturbide et al., 2020), each of which represent consistent regional climate features. The rationale behind their definition, and their homogeneity, are documented in Iturbide et al. (2020). Nine of our regions are continental (albeit with some local surrounding seas); our 'Rest of world' region incorporates the majority of the oceans and the remotest continental areas of Antarctica and Greenland.

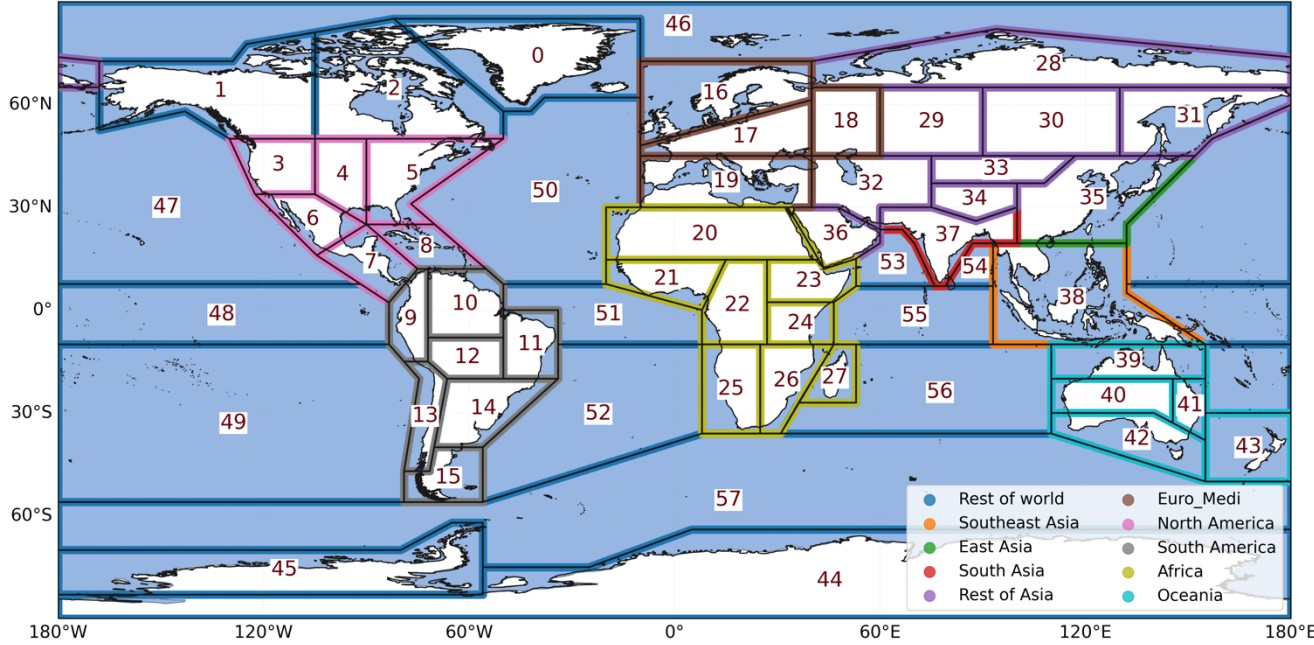

**Figure 1: The boundaries of the 10 world regions used in this study, which in turn are based on the IPCC reference regions described in Iturbide et al. (2020).**

## 3 Results and discussion

### 3.1 Global $N_r$ and $S_r$ concentration distributions

#### 3.1.1 Surface distributions

Figure 2 shows the annual mean surface concentrations of RDN, OXN and OXS and the proportions of each of these that are in the aerosol phase. Maximum concentrations of RDN, OXN and OXS are observed over the continental regions, particularly eastern China and northern India. These are the regions with largest population densities and emissions of $NH_3$, $NO_x$ and $SO_x$, as shown in Ge et al. (2021). The surface distributions of these $N_r$ and $S_r$ species are broadly consistent with previous global modelling studies using GEOS-Chem (Zhu et al., 2015), STOCHEM-CRI (Khan et al., 2020), LMDz-INCA (Hauglustaine et al., 2014) and IMPACT (Xu and Penner, 2012).

The maximum RDN concentrations reach 39 µgN m⁻³ in eastern China, 27 µgN m⁻³ in northern India, 19 µgN m⁻³ in southern Europe, and 7-12 µgN m⁻³ in eastern United States, south-eastern South America, and eastern Africa. Fertilizer application and livestock farming contribute most to $NH_3$ emissions in East Asia, South Asia, Europe, and North America, while biomass burning plays an important role in $NH_3$ emissions in central and eastern Africa (Aksoyoglu et al., 2020; Bray et al., 2018; Dammers et al., 2019; Wang et al., 2018; Whitburn et al., 2015). All these regions combine both high concentrations of RDN and small proportions of $NH_4^+$ (typically <20%). Gaseous $NH_3$ therefore dominates RDN, indicating that whist reducing $NH_3$ emissions over these regions will effectively decrease total RDN concentrations it will have little effect on mitigating SIA formation. A clear increasing gradient of $NH_4^+$ fraction from around 20% to ~80% is observed away from these hotspots indicating conversion of $NH_3$ to $NH_4^+$ as it disperses from emission sources. The longer lifetime and consequent longer travel distance of $NH_4^+$ compared with $NH_3$ (see further discussion on this in Sect. 3.3), means that $NH_4^+$ accounts for over 90% of RDN in remote areas such as over the oceans, deserts, and polar regions.

In general, the concentrations of surface OXN and OXS are also greatest in regions of high RDN concentrations (Fig. 2). This is especially the case over eastern China (maximum: 49 µgN m⁻³ for OXN, 31 µgS m⁻³ for OXS) and northern India (maximum: 33 µgN m⁻³ for OXN, 14 µgS m⁻³ for OXS). The global maximum OXS concentration (54 µgS m⁻³) surrounds Norilsk in northern Russia where there are very large $SO_2$ emissions from nickel and palladium mining and smelting operations

(Fioletov et al., 2020). As illustration of the magnitude of the $SO_x$ emission from that single model grid (560 Gg yr$^{-1}$), it is double the $SO_x$ emission for the whole of the UK (274 Gg yr$^{-1}$) in 2015.

Over continental regions, OXN is predominantly gaseous ($NO_x$, $HNO_3$, HONO, $N_2O_5$, etc.) in areas with both high and low OXN concentrations apart from the remote arid regions of the Sahara, Arabia, and Tibetan Plateau. The high proportion of gaseous OXN is driven by the widespread emissions of $NO_x$ wherever there is population. In contrast, $NO_3^-$ is the major form of OXN over oceans, aside from the major shipping routes (e.g., the sea route from the southern edge of Africa to Southeast Asia) where there are also abundant $NO_x$ emissions. The general dominance of aerosol OXN over the oceans is driven both by sea-salt particles acting as an important marine source for coarse $NO_3^-$ (Lee et al., 2014; Schwier et al., 2017) and the reduced depositional sink for gaseous OXN giving time for gas to aerosol OXN conversion.

The formation of fine $NO_3^-$ and coarse $NO_3^-$ follows different chemical pathways and therefore their concentrations and spatial distributions are closely associated with the differing availability of their individual precursors. Figure S1 shows that fine $NO_3^-$ predominates within total $NO_3^-$ over eastern China, Europe, eastern North America, and south-eastern South America, reflecting a relative abundance of gaseous $NH_3$ in these regions. In contrast, coarse $NO_3^-$ accounts for >80% of total nitrate over India, Africa, and most of the oceans, which indicates large sources of coarse particles in these regions. Karydis et al. (2016) reported similar high contribution of coarse $NO_3^-$ to total $NO_3^-$ over India from their EMAC modelling of global aerosols. Studies of ambient measurements in India also conclude that the relatively high fraction of coarse nitrate is related to high levels of dust and/or sea salt over this region (Kumar et al., 2010; Boreddy et al., 2021).

OXS is globally dominated by the $SO_4^{2-}$ aerosol phase (Fig. 2); $SO_2$ only accounts for more than a few % of OXS in the locality of major $SO_2$ emissions areas such as eastern China, northern Russia, eastern Kazakhstan, and South Africa. This reflects the fast oxidation of $SO_2$ to $SO_4^{2-}$ and that OXS mainly exists in the form of $SO_4^{2-}$ during regional transport.

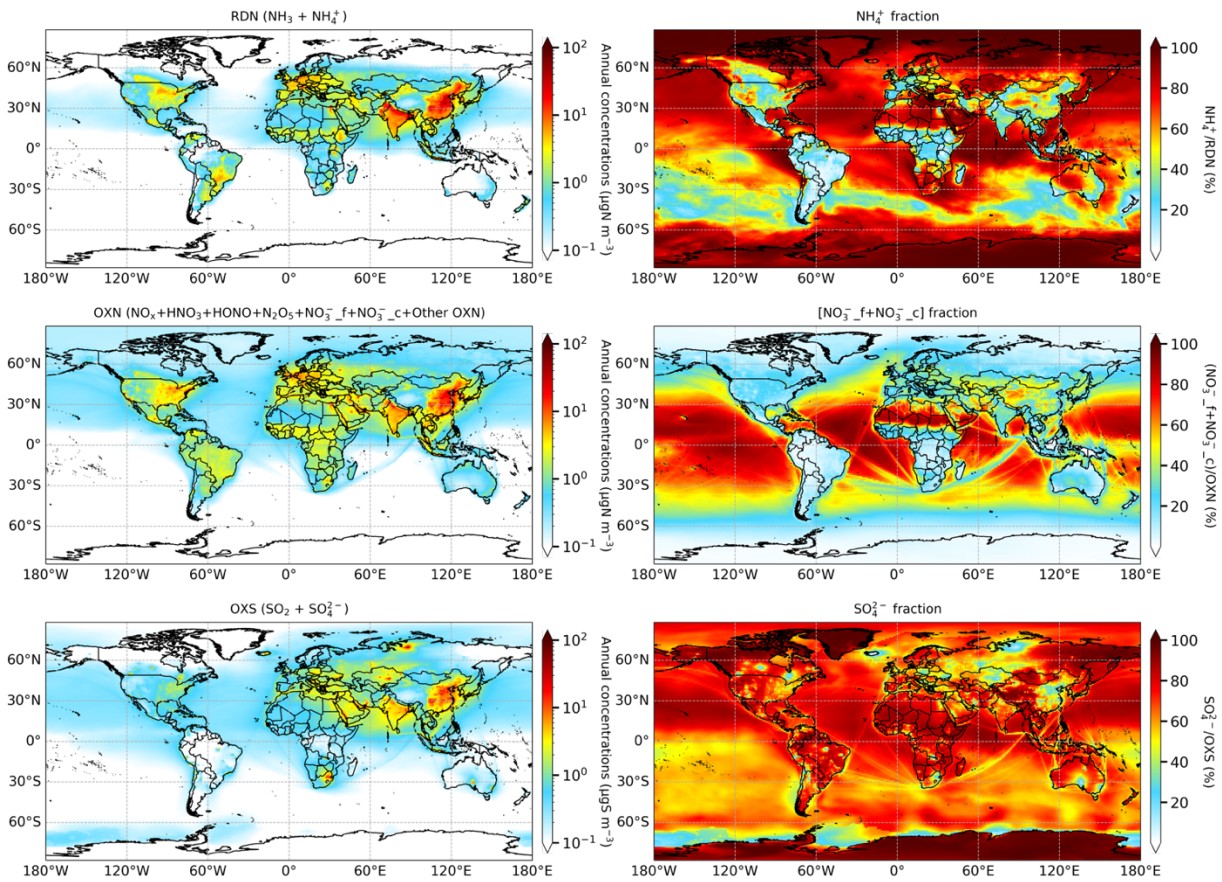

**Figure 2: (Left) Annual mean surface concentrations (log scales) and (right) inorganic aerosol-phase fractions of (top) RDN, (middle) OXN, and (bottom) OXS species. The full composition of OXN is listed in Table S1.**

### 3.1.2 Spatial characterisation of $SO_4^{2-}$-$NO_3^-$-$NH_4^+$ chemical domains

We apply the concept of chemical domains to characterise the state of ammonia neutralisation of both sulfuric and nitric acids (Metzger et al., 2002b; Metzger et al., 2002a; Hauglustaine et al., 2014; Xu and Penner, 2012; Ansari and Pandis, 1998). Figure 3 shows the global spatial distribution in relative abundances of $SO_4^{2-}$, fine $NO_3^-$ and $NH_4^+$ when classified into four such domains. Global maps of the variation in the quantitative species ratios underpinning categorisation into the domains shown in Fig. 3 are presented in Fig. S2a for the ratio of total ammonia $T_A$ to total sulfate $T_S$, and Fig. S2b for the ratio of free ammonia $T_{A\text{-}free}$ to total nitrate (fine) $T_{N\text{-}f}$. The definitions of these four quantities (in molar concentrations; µmol m$^{-3}$) are given in equations E1-E4. The $SO_4^{2-}$ in $T_S$ refers to all of $H_2SO_4$, $NH_4HSO_4$, $(NH_4)_3H(SO_4)_2$, and $(NH_4)_2SO_4$.

$$T_S = [SO_4^{2-}] \qquad\qquad (E1)$$

$$T_{N-f} = [HNO_3] + [NO_3^- \_f] \qquad\qquad (E2)$$

$$T_A = [NH_3] + [NH_4^+] \qquad\qquad (E3)$$

$$T_{A-free} = T_A - 2 \times T_S \qquad\qquad (E4)$$

The $T_{A\text{-}free}$ term quantifies the amount of total ammonia that is in excess of that needed to fully neutralise the sulfate, i.e., the $T_A$ remaining when $\frac{T_A}{T_S} > 2$. When $\frac{T_{A-free}}{T_{N-f}} > 1$, sufficient ammonia exits also to fully react with nitrate to from $NH_4NO_3$. An arbitrary threshold for the sum of the concentrations of fine $SO_4^{2-}$, $NO_3^-$ and $NH_4^+$ to be ≥0.5 µg m$^{-3}$ has been applied to Fig. 3 in order to focus attention away from the most remote areas (southern hemisphere oceans and the poles) where fine SIA concentration is very low and knowledge of chemical domain is irrelevant for informing emissions mitigation strategies.

In the blue and yellow areas of Fig. 3, $\frac{T_A}{T_S} < 2$ and sulfuric acid is not fully neutralised and thus not all $SO_4^{2-}$ is in the form of $NH_4(SO_4)_2$. Locations where this ratio is between 1 and 2 are coloured blue in Fig. 3 (green colours in Fig. S2a and grey areas in Fig. S2b) and are defined as 'sulfate-rich.' This is the state over northern hemisphere oceans and remote continental regions such as Sahara Desert, Greenland, and Siberia. Over northern Russia and southern hemisphere oceans, where $\frac{T_A}{T_S} < 1$ (yellow colour in Fig. 3), conditions can be characterised as 'sulfate-very-rich'. As discussed in Sect. 3.1.1, northern Russia is an $SO_2$ emission hotspot but $NH_3$ emissions are small. This is also the case around other very large point sources of anthropogenic $SO_2$ emissions in Kazakhstan, southern Africa and southern Australia. Over ocean areas, biogenic emission of dimethyl sulphide (DMS) is the major source of sulfur in marine aerosol (Fiddes et al., 2018; Jackson et al., 2020) but $NH_3$ emission is again very small. A separation between sulfate-rich and sulfate-very-rich regions arises because in the former areas there is transport of anthropogenic continental $NH_3$ emissions over the northern hemisphere oceans. In these sulfate-rich and sulfate-very-rich regions essentially all $NH_3$ has been converted to $NH_4^+$ as illustrated in Fig. 2 where RDN concentrations are almost entirely in the form of $NH_4^+$.

The areas with green and orange colours in Fig. 3 (yellow to red colours in Fig. S2a) have $\frac{T_A}{T_S} > 2$ and consequently $\frac{T_{A-free}}{T_{N-f}} > 0$ (non-grey colours in Fig. S2b) and show where sulfuric acid is fully neutralised and excess $NH_3$ remains to react with $HNO_3$ to form $NH_4NO_3$. Most continental regions with significant SIA pollution are characterised by these conditions including East Asia, Southeast Asia, South Asia, Europe, North America, south-eastern South America, western Africa, and eastern Australia.

The green coloured areas in Fig. 3 have a $\frac{T_{A-free}}{T_{N-f}}$ ratio between 0 and 1 and indicate where $NH_3$ is the limiting factor for $NH_4NO_3$ formation and thus where the chemical domain can be characterised as '$NO_3^-$ rich.' The blue colours in Fig. S2b illustrate the quantitative extent of the $HNO_3$ neutralisation in these areas. This situation occurs over coastal areas around East Asia, northern Europe, and north-eastern United States. These locations experience transport of oxidised N from the adjacent continent and/or direct shipping emissions of $NO_x$ but $NH_3$ concentrations are lower because it deposits more quickly to land before transport off the coast.

The orange-coloured areas in Fig. 3 are where $\frac{T_{A-free}}{T_{N-f}} > 1$ and are characterised as 'NH$_3$ very rich', since here it is the availability of nitrate that limits NH$_4$NO$_3$ formation. This condition prevails over most of the temperate and tropical land surface in the northern hemisphere. Concentrations of SIA over a lot of the land in the southern hemisphere are (like over the southern hemisphere oceans) less than the 0.5 µg m$^{-3}$ threshold applied to Fig. 3. Figure S2b shows that the most NH$_3$-rich areas are in Central Africa, South Asia, East Asia, Southeast Asia, western Europe, South and Central America, North America,

and New Zealand where there are intensive agricultural activities and/or large biomass burning and/or low NO$_x$ and SO$_x$ emissions.

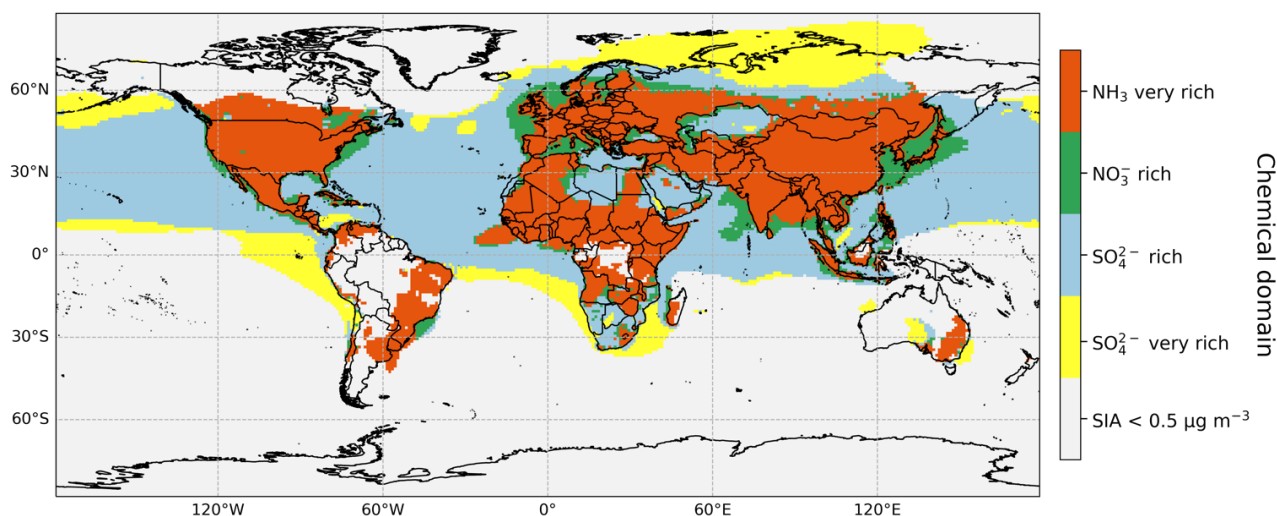

**Figure 3: The state of ammonia neutralisation of sulfuric and nitric acids for areas where SIA concentration in the fine particle**
**fraction (PM$_{2.5}$) is ≥0.5 µg m$^{-3}$. Definitions of the four chemical domains are given in the main text. Maps of the quantitative species ratios underpinning categorisation into these chemical domains are given in Fig. S2.**

        Hauglustaine et al. (2014) and Xu and Penner (2012) reported similar findings, although both these studies also showed NH$_3$-very-rich conditions over tropical oceans, which may be related to oceanic NH$_3$ emissions if these were included in their

simulations. They were not included in our model. Oceanic NH$_3$ emissions are small (~6 TgN yr$^{-1}$, albeit highly uncertain (Fowler et al., 2015)), compared to continental NH$_3$ emission (53.1 TgN yr$^{-1}$; Table 1) and we deliberately exclude discussion of chemical domain where SIA concentration is small.

        The continental values of $\frac{T_{A-free}}{T_{N-f}}$ calculated here for 2015 are larger than those reported for 2000. For instance, in western Europe, Xu and Penner (2012) and Hauglustaine et al. (2014) reported $\frac{T_{A-free}}{T_{N-f}}$ ratios of around 1.0-2.0 and 1.0-4.0, respectively,

00    while we report values around 4.0-10.0 (Fig. S2b). This temporal difference is driven not only by the increasing NH$_3$ emissions associated with sustaining food supply for the growing global population (Sutton et al., 2013; Behera et al., 2013b; Aneja et al., 2009), but also because a growing number of countries have been reducing NO$_x$ and SO$_x$ emissions yet most countries have little or no control policy on NH$_3$ emissions (Liu et al., 2018; Liu and Matsui, 2021; Aas et al., 2019; de Gouw et al., 2014). These changes in relative emissions of precursors mean an increasingly smaller proportion of gaseous NH$_3$ is transformed to

05    aerosol NH$_4^+$, which in turn increases NH$_3$ concentrations and reduces the effectiveness of PM$_{2.5}$ mitigation via NH$_3$ emission reductions. For example, Liu et al. (2018) reported a significant growth in gaseous NH$_3$ concentrations over the North China Plain from 2008 to 2016 allied to the rapid SO$_2$ emission reductions.

        Our results also indicate regional differences in terms of the extent of excess ammonia (Fig. S2b). Table S3 lists the maximum, median and area-weighted mean $\frac{T_{A-free}}{T_{N-f}}$ ratios in different world regions. Among the regions with largest RDN

surface concentration, South Asia shows the most 'NH₃-rich' condition with an area-weighted mean ratio of 9.6, followed by South America and Africa whose area-weighted mean ratios are 8.6 and 7.2 respectively. For regions with area-weighted mean $\frac{T_{A-free}}{T_{N-f}}$ ratio > 1, Southeast Asia and Euro_Medi have the lowest area-weighted mean ratios of 1.9 and 2.5, respectively. This lower relative excess of NH₃ of the former is associated with the large proportion of marine area in this region, while the smaller NH₃ abundance in the latter is likely linked to more stringent NH₃ emission targets in Europe than other world regions via the Gothenburg Protocol and EU National Emissions Ceiling Directive.

## 3.2 Global and regional $N_r$ and $S_r$ deposition

An important advantage of an atmospheric model is that it can apportion the different contributions to deposition that measurements cannot readily quantify. For instance, direct measurements of dry deposition are technically difficult and are very limited in terms of both spatial and temporal coverage. Also, whilst measurement of NH₄⁺ in precipitation permits quantification of wet deposition of RDN, it cannot distinguish how much of it derives from rainout of gaseous NH₃ and how much from rainout of particle NH₄⁺. Similarly, measurements cannot distinguish between gaseous and particle sources to wet deposition of OXN and OXS. The models therefore provide additional information about the nature of $N_r$ and $S_r$ deposition. This is relevant as the form of deposited N affects the severity of nitrogen deposition impacts (Sutton et al., 2014; Bobbink et al., 2010). An important effect of deposited N gases and aerosols is the direct toxicity to above-ground parts of various plants. Van Herk et al. (2003) found that lichens are particularly sensitive to direct NH₃ dry deposition, whilst Pescott et al. (2015) reported direct toxic effects of wet deposited RDN for bryophytes. Additionally, it must be stressed that the lack of dry deposition comparison between model and measurement, due to the lack of measurement data, means that the numbers reported for modelled dry deposition in this section should be regarded as more uncertain.

### 3.2.1 RDN

Figure 4 shows the global distribution of annual RDN deposition in 2015 apportioned into dry and wet deposition of each of NH₃ and NH₄⁺. The map of total RDN deposition (the sum of all panels of Fig. 4) is provided in Fig. S3. The relative contributions of the different sources of RDN deposition are illustrated in Fig. 5 for the 10 world regions outlined in Fig. 1. The four panels of Fig. 4 show similar spatial patterns of highest depositions over East Asia, South Asia, Europe, eastern North America, south-eastern South America, and central Africa. The greatest RDN deposition (wet + dry) rate occurs in East Asia, exceeding 5300 mgN m⁻² yr⁻¹. The East Asia region receives more than one-fifth of the world's RDN deposition (Fig. 5). Maximum total RDN deposition in South Asia and Southeast Asia is in the range 3800 to 4600 mgN m⁻² yr⁻¹, whilst in Europe, North America, South America, and Africa maximum total RDN deposition is in the range 1800 to 2200 mgN m⁻² yr⁻¹ (Fig. S3). These regions each receive 7-11% of global total RDN deposition (Fig. 5). Around 15% of RDN deposition occurs over the oceans ('Rest of World in Fig. 5). In fact, since most of the other world regions also include areas of water, the proportion of global RDN depositing to seas will be greater than 15%.

Globally, the total RDN deposition is 55.5 TgN yr⁻¹, apportioned around 50% each between dry and wet deposition. The global deposition can be further apportioned as 40% NH₃ dry deposition, 9% NH₄⁺ dry deposition, 22% wet deposition of NH₃ and 28% wet deposition of NH₄⁺ (Table S4). Thus, whilst the large majority (82%) of RDN dry deposition is in the form of NH₃, the RDN wet deposition is more evenly split between rainout of gaseous NH₃ (44%) and rainout of particle NH₄⁺ (56%). Overall, 62% and 38% of total global RDN is removed from the atmosphere initially in the form of NH₃ and NH₄⁺, respectively.

However, Fig. 5 shows there are large regional variations in the contributions of different sources of RDN deposition. In South Asia, South America, and Oceania more than 50% of RDN deposition is in the form of NH₃ dry deposition. In East Asia, Euro_Medi, North America, and Africa dry deposition of NH₃ is in the range 42-49%. Wet deposition of NH₄⁺ is

dominant only in remote areas such as Rest of world (54%) and Rest of Asia (40%). Overall, the deposition of NH₃ (dry + wet) accounts for well over half of total RDN deposition in all continental regions (except for Rest of Asia), which indicates that reducing NH₃ emissions will be effective at reducing total RDN input from the atmosphere to continental ecosystems.

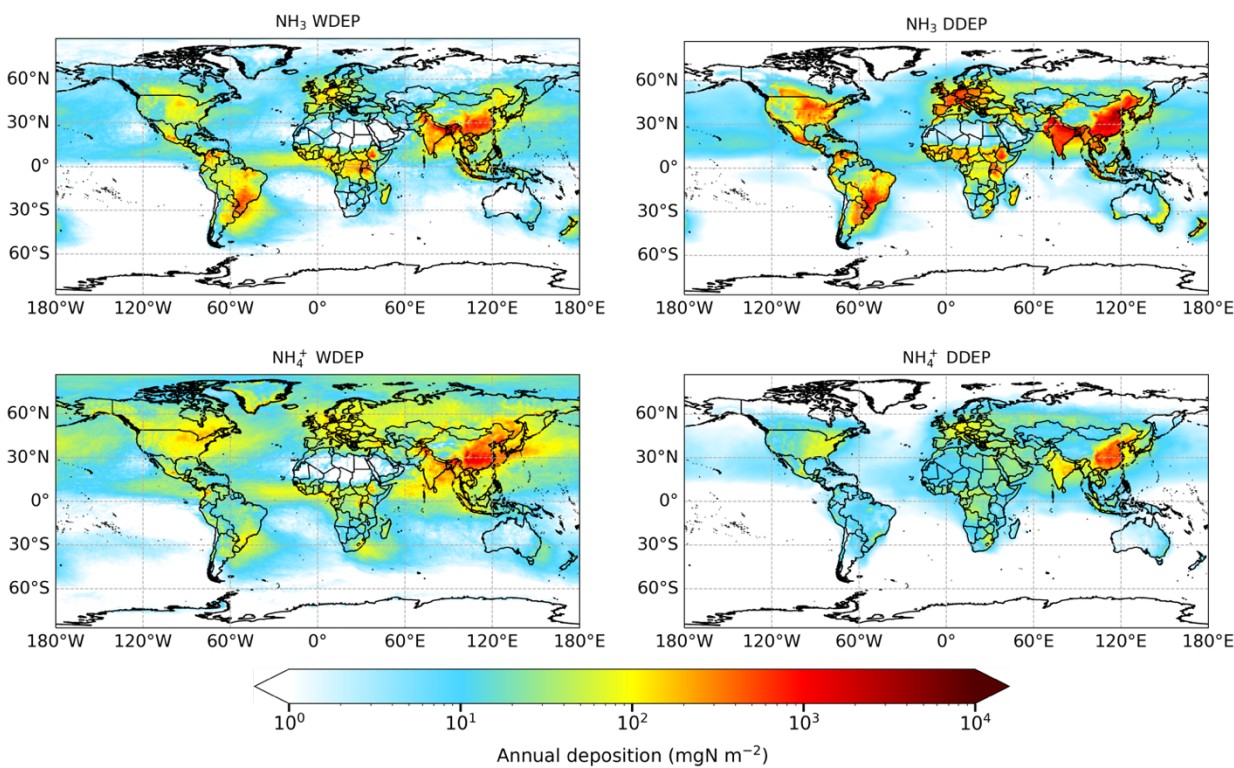

**Figure 4: Annual wet (WDEP) and dry deposition (DDEP) of RDN components for 2015. 'NH₃ WDEP' and 'NH₄⁺ WDEP' respectively refer to wet deposition of RDN originally present in the atmosphere as gaseous NH₃ or as aerosol NH₄⁺. Note the logarithmic scale.**

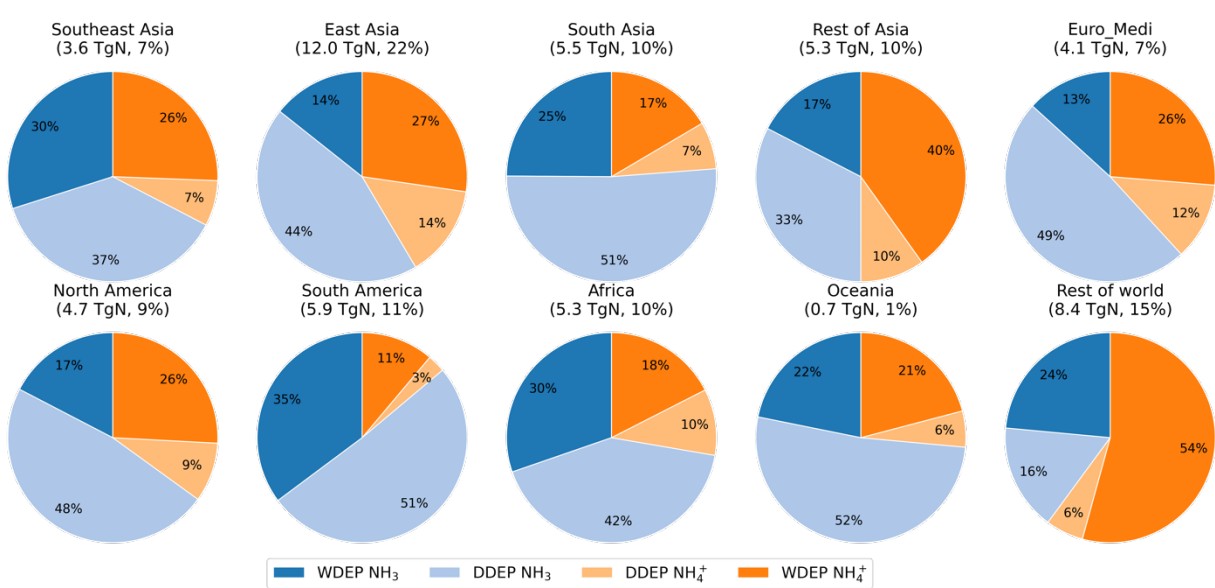

**Figure 5: Relative contributions to annual RDN deposition for the set of world regions defined in Fig. 1. The two darker shade colours collectively indicate wet deposition (WDEP), whilst the two lighter shade colours collectively indicate dry deposition (DDEP). The total deposited RDN (TgN) in each region and its proportion to global total RDN deposition are presented below each region name.**

**3.2.2 OXN**

Figure 6 shows the global distribution of annual OXN deposition in 2015 apportioned into dry and wet deposition of gaseous OXN (i.e., $NO_x$ + $HNO_3$ + HONO + $N_2O_5$ + other OXN species) and aerosol-phase OXN ($NO_3^-$_f + $NO_3^-$_c). The map of total OXN deposition is in Fig. S4. As described above for RDN deposition, model simulations are able to distinguish in detail the ultimate atmospheric source of the OXN deposition, which measurements cannot; in particular to distinguish whether dissolved nitrate in precipitation derives from gas or aerosol-phase species, and which species. The relative contributions of the different sources of OXN deposition for the 10 world regions outlined in Fig. 1 are shown in Fig. 7.

The regions of high OXN deposition in Fig. 6 are similar to those with high RDN deposition (Fig. 4), although shipping $NO_x$ emissions can be seen to have a notable impact on deposition of OXN over a large number of marine locations, which is not apparent for RDN deposition. The largest amounts of total OXN deposition are in East Asia (3300 mgN $m^{-2}$ $yr^{-1}$), followed by Southeast Asia (2800 mgN $m^{-2}$ $yr^{-1}$) and South Asia (1700 mgN $m^{-2}$ $yr^{-1}$) (Fig. S4). However, whilst the East Asia region receives 22% of global RDN deposition (Fig. 5), this region receives only 14% of global OXN deposition (Fig. 7). In North America, maximum OXN deposition is around 1900 mgN $m^{-2}$ $yr^{-1}$, whilst in Europe it is around 900 mgN $m^{-2}$ $yr^{-1}$. A greater proportion of OXN deposition occurs over the oceans than is the case for RDN deposition ('Rest of World' proportions are 23% and 15%, respectively), due both to the shipping emissions of $NO_x$ and the longer lifetime (and hence further transport distance) of $NO_2$ compared with $NH_3$.

The global total OXN deposition in 2015 is 57.9 TgN $yr^{-1}$, apportioned as 43% dry deposition and 57% wet deposition. The relative contributions of the individual OXN components to global OXN deposition is shown in Fig. S6. The global deposition is further apportioned as 28% gaseous OXN dry deposition, 15% aerosol-phase OXN dry deposition, 18% wet deposition of gaseous OXN and 39% wet deposition of aerosol-phase OXN (Table S4). The global OXN dry deposition is almost equally split among $HNO_3$ (34%), coarse $NO_3^-$ (29%), and $NO_x$ (21%, this is effectively dry deposition of $NO_2$), whereas the rainout of coarse particle $NO_3^-$ (60%) is the predominant form in OXN wet deposition, followed by the rainout of $HNO_3$ (28%). The dry-deposited fine $NO_3^-$ and the rainout of fine $NO_3^-$ only account for 7% and 9% of OXN dry deposition and OXN wet deposition, respectively. The total deposition (wet + dry) of 'Other' OXN species (predominantly PAN) contributes 5% to global total OXN deposition, and the contributions from HONO and $N_2O_5$ are negligible. Compared with global RDN deposition, the majority of which derives from wet and dry deposition of gaseous $NH_3$ (62%), Fig. S6 shows it is coarse particle $NO_3^-$ (46%) that contributes most to global OXN deposition, followed by $HNO_3$ (30%). Overall, slightly greater OXN deposition derives from aerosol species (54%) than from gaseous species (46%) globally.

Figure 7 shows that the 10 world regions can be viewed in four groups according to the nature of their OXN deposition. A group comprising Southeast Asia, South America and Oceania is characterised by large contributions to total OXN deposition derived from wet and dry deposition of gaseous species. In Southeast Asia, wet and dry deposition of $HNO_3$ constitutes over half (52%) of total OXN deposition, with coarse $NO_3^-$ (31%) and $NO_x$ (7%) being the second and third contributors. This indicates that it is the formation of gaseous species (mainly $HNO_3$ and $NO_2$) that are the most important determinants of OXN deposition in this region. Similar observations are made for South America and Oceania. Fine $NO_3^-$ contributes very little to total OXN deposition in this group of regions.

In the regions of South Asia, Rest of Asia, Africa, and Rest of world, it is coarse $NO_3^-$ (via both wet and dry deposition) that is the most significant contributor to total OXN deposition (wet + dry). This is consistent with the relatively greater abundance of coarse particle $NO_3^-$ in these regions (Fig. S1).

A third group of regions from Fig. 7 comprises North America and Euro_Medi. In these regions, the proportion contributions to total OXN deposition from wet and dry deposition of $HNO_3$ and coarse $NO_3^-$ are similar at 31-38% and there is also significant contribution from dry deposition of $NO_x$ (13-14%) and 'Other OXN' (6% and 3% for North America and Euro_Medi, respectively). The relative contributions from $NO_x$ in these regions are well above their average contribution

globally. There is also an important contribution (8-10%) from wet and dry deposition of fine $NO_3^-$ to total OXN deposition in these two regions.

Finally, the East Asia region is characterised by substantial contribution (24%) to total OXN deposition from wet and dry deposition of fine $NO_3^-$, a proportion that is more than double the fine $NO_3^-$ contribution in North America and Euro_Medi. The formation of $NH_4NO_3$ aerosol is therefore an important influence on OXN deposition over East Asia, in addition to its important contribution to $PM_{2.5}$ and to the latter's health effects. The atmospheric components $HNO_3$ and coarse $NO_3^-$ contribute equal amount (30%) to total wet and dry OXN deposition in East Asia.

Globally, our modelling shows that OXN (51%) and RDN (49%) account for almost equal proportions of the total nitrogen deposition (113.4 TgN yr$^{-1}$). Other studies report similar values with OXN and RDN deposition amounts each varying in the range 50 to 65 TgN yr$^{-1}$ and contributing comparable amounts to total nitrogen deposition (Hauglustaine et al., 2014; Lamarque et al., 2013; Dentener et al., 2006; Bian et al., 2017; Xu and Penner, 2012; Tan et al., 2018).

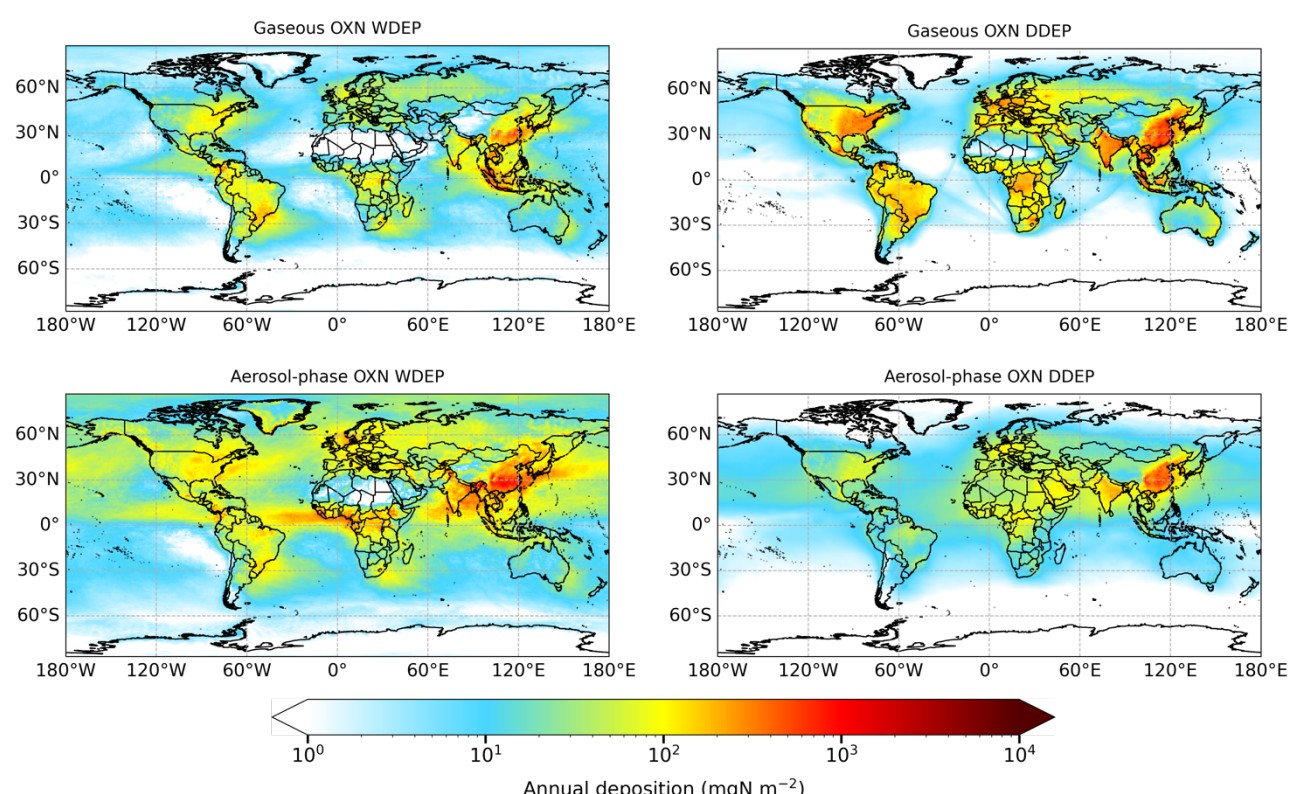

**Figure 6: Annual wet (WDEP) and dry deposition (DDEP) of OXN components for 2015. 'Gaseous OXN WDEP' and 'Aerosol-phase OXN WDEP' respectively refer to wet deposition of OXN originally present in the atmosphere as gaseous OXN or as aerosol-phase OXN. The former consists of $NO_x$, $HNO_3$, HONO, $N_2O_5$ and other OXN (predominantly PAN), whilst the latter consists of $NO_3^-$_f and $NO_3^-$_c (full compositional details in Table S1). Note the logarithmic scale.**

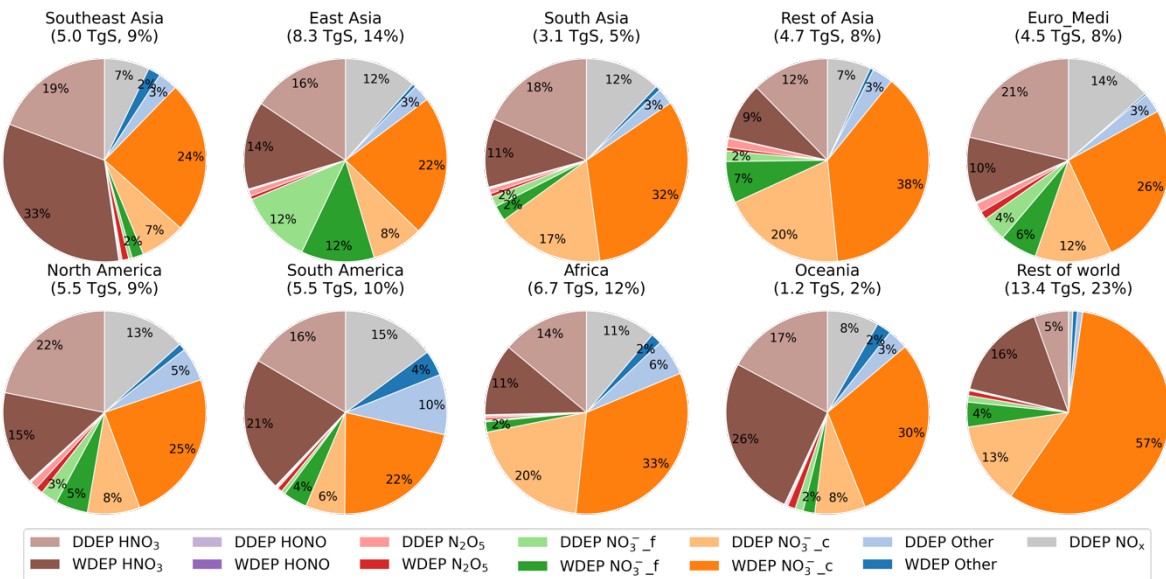

**Figure 7: Relative contributions to annual OXN deposition for the set of world regions defined in Fig. 1. Labels for individual contributions <2% are omitted to aid clarity. The darker shade colours collectively indicate wet deposition (WDEP), whilst the lighter shade colours collectively indicate dry deposition (DDEP). The 'Other' OXN species is predominantly PAN (full compositional details in Table S1). The total deposited OXN (TgN) in each region and its proportion to global total OXN deposition are presented below each region name.**

### 3.2.3 OXS

Figure 8 shows the global distribution of annual OXS deposition apportioned into wet and dry deposition of $SO_2$ and $SO_4^{2-}$. The map of total OXS deposition is provided in Fig. S5. The relative contributions of the different sources of OXS deposition for the 10 world regions outlined in Fig. 1 are shown in Fig. 9. Higher levels of OXS deposition are found near emission sources in East Asia, South Asia, Europe, eastern United States, and south-eastern Africa for all four deposition components. In addition, both wet and dry deposition of $SO_2$ is large close to major metal smelting industry such as in northern Russia, where total OXS deposition exceeds 5400 mgS m$^{-2}$ yr$^{-1}$, eastern Kazakhstan, and South Africa. In East Asia and South Asia, the highest total (wet + dry) OXS depositions are around 4800 mgS m$^{-2}$ yr$^{-1}$ and 2400 mgS m$^{-2}$ yr$^{-1}$, respectively. OXS deposition in Europe and North America is much smaller, with largest values in the range 900 to 1100 mgS m$^{-2}$ yr$^{-1}$. A substantial proportion of total OXS deposition occurs over the oceans; 34% occurs over the Rest of world region (Fig. 9), which reflects both deposition of S derived from oxidation of natural DMS emissions, and the longer transport distances of anthropogenic aerosol $SO_4^{2-}$. Of the continental regions, Fig. 9 shows the greatest proportions of OXS deposition occur in East Asia (16%) and Rest of Asia (11%).

Globally, total OXS deposition in 2015 is 50.5 TgS yr$^{-1}$, of which 54% is via wet deposition and 46% via dry deposition. The largest contributions to total OXS deposition are wet deposition of aerosol $SO_4^{2-}$ (44%) and dry deposition of $SO_2$ (35%) (Table S4). Rainout of $SO_2$ into wet deposition contributes 11% and dry deposition of aerosol $SO_4^{2-}$ contributes 10%. This means that, globally, slightly greater OXS deposition derives from aerosol $SO_4^{2-}$ (54%) than from $SO_2$ (46%). This is a similar situation to that for OXN deposition but contrasts with RDN deposition for which 62% ultimately derives from the gaseous component ($NH_3$) and only 38% from the aerosol component ($NH_4^+$).

The total OXS deposition we calculate for 2015 is around half that reported by Hauglustaine et al. (2014) for 2000 (107 TgS yr$^{-1}$) and 40% lower than that reported by Tan et al. (2018) for 2010 (84 TgS yr$^{-1}$). It is also lower than the values for 2000 (range 75-100 TgS yr$^{-1}$) from a 23 model intercomparison reported by Dentener et al. (2006). The lower OXS deposition value presented here is assumed in part to be due to the considerable reduction in global sulfur emission from 2000 to 2015 (Fowler

et al., 2020; Aas et al., 2019). A direct comparison of modelled S deposition between our work and other studies is confounded by use of different inventories for both anthropogenic and natural emissions.

As for RDN and OXN, the relative contributions of the different sources of OXS deposition vary across the 10 world regions (Fig. 9). However, the regional composition patterns of OXS deposition are broadly similar to those of RDN deposition, with dry deposition of $SO_2$ (23-46%) and wet deposition of $SO_4^{2-}$ (34-56%) being the two most important contributors in all world regions. Total deposition (wet + dry) of $SO_2$ and $SO_4^{2-}$ contribute almost equal proportions to total OXS deposition in Southeast Asia, South Asia, North America, South America, and Oceania. In East Asia and Euro_Medi regions, the sum of $SO_2$ deposition contributes slightly more (55-56%) to total OXS deposition, whereas it is $SO_4^{2-}$ that contributes most (57-67%) to total OXS deposition in Rest of Asia, Africa, and Rest of world.

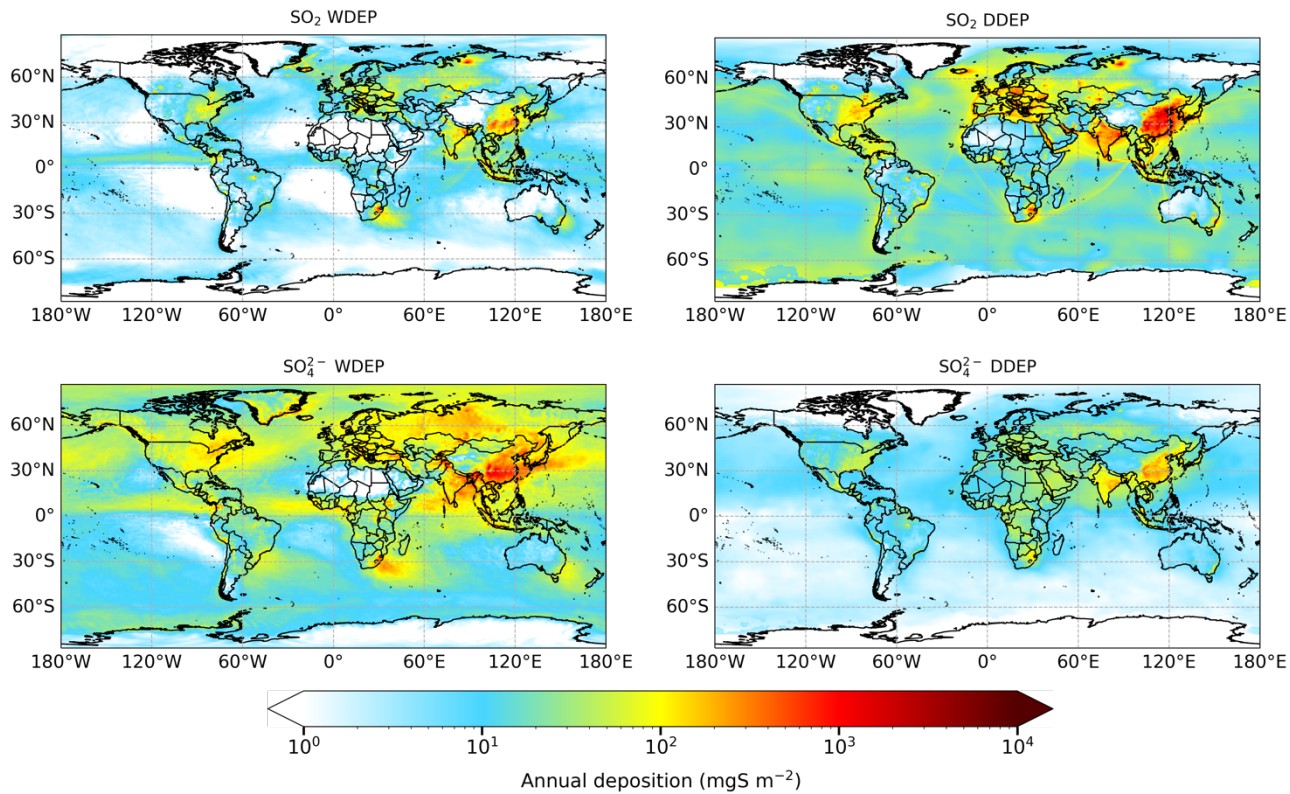

**Figure 8: Annual wet (WDEP) and dry deposition (DDEP) of OXS components for 2015. 'SO₂ WDEP' and 'SO₄²⁻ WDEP' respectively refer to wet deposition of OXS originally present in the atmosphere as gaseous SO₂ or as aerosol SO₄²⁻. Note the logarithmic scale.**

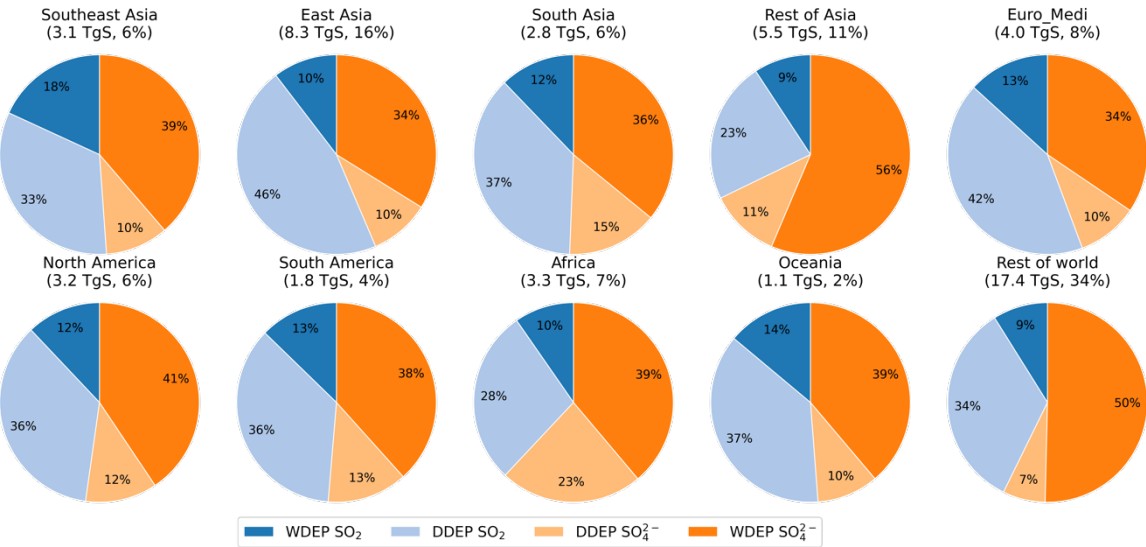

**Figure 9: Relative contributions to annual OXS deposition for the set of world regions defined in Fig. 1. The two darker shade colours collectively indicate wet deposition, whilst the two lighter shade colours collectively indicate dry deposition. The total deposited OXS (TgS) in each region and its proportion to global total OXS deposition are presented below each region name.**

In summary, on a global basis, wet and dry deposition currently contribute equally to RDN deposition. For OXN and OXS, wet deposition contributes slightly more than dry deposition. These proportions vary regionally according to local precursor emission strengths and climatology. For RDN deposition, dry deposition of $NH_3$ is the predominant contributor in most regions except for Rest of Asia and Rest of world (essentially oceans) where $NH_3$ emissions are small and RDN deposition is largely determined by transport of aerosol $NH_4^+$ from other regions (see Sect. 3.2.1). Further reductions in $NH_3$ would thus efficiently reduce the deposition of RDN in most world regions. The two largest contributors to OXN deposition in all regions are $HNO_3$ and coarse $NO_3^-$ (both occurring by both wet and dry deposition). Dry deposition of $NO_x$ (i.e., $NO_2$) is the third largest contributor in most regions except for East Asia and Rest of world. The deposition of fine $NO_3^-$ is only important over East Asia, and to a lesser extent in Euro_Medi. Due to the general dominance of aerosol-phase $NO_3^-$ in OXN deposition, changes in primary $NO_x$ emissions may not have a direct and linear influence on the deposition of OXN. Sensitivity experiments focusing on the response of deposition to emission changes are required to gain more insight into this issue. For OXS deposition, $SO_2$ and $SO_4^{2-}$ contribute almost equally to total OXS deposition in most regions. More specifically, the sum of dry deposition of $SO_2$ and wet deposition of $SO_4^{2-}$ accounts for at least 67% of total OXS deposition in all world regions.

## 3.3 Global and regional $N_r$ and $S_r$ budgets

### 3.3.1 Global budgets

Table 1 summarizes global tropospheric budget data for RDN, OXN and OXS for 2015. Table 2 does likewise for the individual components $NH_3$, $NH_4^+$, $HNO_3$, fine $NO_3^-$, coarse $NO_3^-$, and $SO_4^{2-}$. In this study, the top of the troposphere is defined as the top of our model domain at 100 hPa. Table 1 shows that net exchange of chemical species at the top of the model domain ('Flux in' and 'Flux out') is negligible. We calculate atmospheric lifetimes based on both sources and sinks. Detailed calculation procedure is described in Supplement Material Sect. S1.2.

The only source of atmospheric RDN is the annual $NH_3$ surface emission of 53.1 TgN. The annual average burden of RDN is 0.75 TgN, and it has a global lifetime of 4.9-5.2 days depending on how this is calculated (Table 1). The total deposition of $NH_3$ is 34.6 TgN yr$^{-1}$ (Table 2). This contributes 62% of the sink of $NH_3$ and occurs as 65% dry deposition and 35% wet deposition. Partitioning of $NH_3$ into $NH_4^+$ aerosol, which is also the only source of $NH_4^+$, contributes 21.0 TgN yr$^{-1}$ (38%) to

the total sink of $NH_3$. The $NH_4^+$ loss comes mainly from wet deposition, 15.8 TgN yr$^{-1}$ (76%), and to a lesser extent from dry deposition, 5.02 TgN yr$^{-1}$ (24%). Net loss to the stratosphere is negligible. The global average tropospheric lifetime of $NH_4^+$ (8.9 days) is much longer than that of $NH_3$ (1.6 days), consistent with the annual average $NH_4^+$ burden (0.51 TgN) being more than double that of $NH_3$ (0.24 TgN) (Table 2).

Xu and Penner (2012), Hauglustaine et al. (2014) and Bian et al. (2017) report similar global total $NH_3$ emissions (50.5-53.6 TgN yr$^{-1}$ for 2000, and 51.8 TgN yr$^{-1}$ for 2008), but smaller burdens and shorter lifetimes of $NH_3$ and $NH_4^+$ than we report here. The burdens of $NH_3$ and $NH_4^+$ are 0.07 TgN (lifetime = 0.46 days) and 0.26 TgN (lifetime = 3.23 days) in Xu and Penner (2012), and 0.09 TgN (lifetime = 0.63 days) and 0.22 TgN (lifetime = 4.52 days) in Hauglustaine et al. (2014) for 2000. Burdens of $NH_3$ and $NH_4^+$ averaged across 9 models in Bian et al. (2017) for 2008 are 0.16 TgN and 0.25 TgN with a corresponding lifetime of 0.72 days and 4.3 days. It is important to note that the $NH_3$ burden across the 9 models in Bian et al. (2017) varied by a factor of 17. This indicates the sensitivity of burdens and lifetimes of $NH_3$ and $NH_4^+$ to the definition of model domain, the deposition pathways, and the interactions between $NH_3$ and acidic species in different models.

Table 1: Global tropospheric budgets of total RDN, OXN, and OXS in 2015. Flux (out - in) = Flux out - Flux in. *Q* is the annual average burden, *P* is the production pathway (i.e., sources), *R* is the removal pathway (i.e., sinks), and *τ* is atmospheric lifetime. The constituents of the three groups are listed in Table S1.

| | Total RDN | Total OXN | Total OXS |
|---|---|---|---|
| Sources (TgN/TgS yr$^{-1}$) | 53.1 | 56.6 | 48.5 |
| Sinks (TgN/TgS yr$^{-1}$) | 55.5 | 57.9 | 50.5 |
| Dry deposition | 27.4 | 24.7 | 23.1 |
| Wet deposition | 28.1 | 33.2 | 27.4 |
| Chemical loss | - | - | - |
| Flux (out - in) | 0.01 | 0.01 | -0.04 |
| Burden (TgN/TgS) | 0.75 | 1.20 | 0.78 |
| Lifetime (days): $\tau = \dfrac{Burden}{Sources}$ | $\dfrac{Q(NH_3 + NH_4^+)}{P(NH_3)}$ <br> 5.2 | $\dfrac{Q(OXN)}{P(NO_X)}$ <br> 7.7 | $\dfrac{Q(SO_2 + SO_4^{2-})}{P(SO_2)}$ <br> 5.9 |
| Lifetime (days): $\tau = \dfrac{Burden}{Sinks}$ | $\dfrac{Q(NH_3 + NH_4^+)}{R(NH_3 + NH_4^+)}$ <br> 4.9 | $\dfrac{Q(OXN)}{R(OXN)}$ <br> 7.6 | $\dfrac{Q(SO_2 + SO_4^{2-})}{R(SO_2 + SO_4^{2-})}$ <br> 5.6 |

The global atmospheric OXN source and sink flux is 56.6 and 57.9 TgN yr$^{-1}$ (Table 1). The annual average OXN burden is 1.20 TgN, with a global mean lifetime of 7.6 or 7.7 days depending on how this is calculated. Detailed global budgets of $HNO_3$ and fine and coarse $NO_3^-$ are listed in Table 2. As quantification of the flux of $HNO_3$ through photolysis and reaction with OH is not available as model output, the $HNO_3$ budget presented here includes only deposition and aerosol nitrate formation. Xu and Penner (2012) and Hauglustaine et al. (2014) report that 6.9%-10% of $HNO_3$ loss occurs via photolysis and reaction with OH, so these loss routes are relatively small. The total chemical production of $HNO_3$ in our study is 49.5 TgN yr$^{-1}$, which is 17% higher than the 42.3 TgN yr$^{-1}$ reported by Xu and Penner (2012), and 2% higher than the 48.5 TgN yr$^{-1}$ in

Hauglustaine et al. (2014). The total loss of $HNO_3$ is 49.4 TgN yr$^{-1}$, of which 17.6 TgN yr$^{-1}$ (36%) is removed from the atmosphere by deposition, almost equally via dry deposition (8.35 TgN yr$^{-1}$) and via wet deposition (9.25 TgN yr$^{-1}$). The remaining 64% of the $HNO_3$ loss goes into formation of 4.54 TgN yr$^{-1}$ fine nitrate (14% of the aerosol nitrate source) and 27.2 TgN yr$^{-1}$ coarse nitrate (86% of the aerosol nitrate source). The tropospheric annual average burden of $HNO_3$ is 0.15 TgN with a lifetime of 1.1 days.

The loss of fine and coarse nitrate is only via deposition, and 72% of total nitrate loss is due to wet deposition. The calculated nitrate lifetimes range between 4.8 and 7.1 days, which are in line with lifetimes (4.6-6.3 days) reported in other studies (Xu and Penner, 2012; Hauglustaine et al., 2014; Bian et al., 2017). Coarse nitrate (7.0-7.1 days) has slightly longer lifetime than fine nitrate (4.8 days). Our calculated total nitrate annual average burden is 0.58 TgN with fine $NO_3^-$ comprising only 10% of this total. This total nitrate burden is around three times greater than the value in Xu and Penner (2012) and Hauglustaine et al. (2014) for 2000 (0.17-0.18 TgN), which likely reflects a shift in gas-aerosol partitioning between 2000 and 2015 from gas-phase $HNO_3$ more towards aerosol-phase $NO_3^-$, especially given that their $HNO_3$ burden of 0.30 TgN is double our value. It is also possible that the simulation of nitrate burdens is more sensitive to coarse particle emissions and the production of coarse nitrate in our model than in models used by Xu and Penner (2012) and Hauglustaine et al. (2014).

The dominance of coarse nitrate in global total nitrate burden is also observed in other global modelling studies. Hauglustaine et al. (2014), Xu and Penner (2012), and Bauer et al. (2007) reported a relative contribution of coarse-mode nitrate of 72%, 77%, and 79% to total nitrate burden in their respective models for 2000. Additionally, the lifetime of fine nitrate calculated in our work is comparable with other literature values. Park et al. (2004) reported a global fine nitrate lifetime against deposition of 3.2 days for 2001, Feng et al. (2007) reported a global lifetime of total nitrate (fine + coarse) of 5.0 days for 1997, and Alexander et al. (2020) reported a global lifetime of inorganic nitrate ($HNO_3$ + total $NO_3^-$) for the period 2000-2015 on the order of 3–4 days.

For OXS, emission of $SO_2$ is the only source, and deposition (wet + dry) is almost the only sink pathway (Table 1). The annual average burden of OXS is 0.78 TgS with a lifetime of 5.6-5.9 days depending on how it is calculated. The total source of $SO_4^{2-}$ is 27.6 TgS yr$^{-1}$ (Table 2). The dominant loss of $SO_4^{2-}$ is wet deposition (81%) with the remaining loss (19%) being dry deposition. The annual average $SO_4^{2-}$ burden in the troposphere is 0.63 TgS. $SO_4^{2-}$ lifetime of 8.3-8.5 days is very similar to the $NH_4^+$ lifetime, which can be explained by the close chemical association between these two species in atmospheric SIA.

**Table 2: Global tropospheric budgets of $NH_3$, $NH_4^+$, $HNO_3$, $NO_3^-$ and $SO_4^{2-}$ in 2015. Flux (out - in) = Flux out - Flux in. $Q$ is the quantity of annual average burden, $P$ is the production pathway (i.e., sources), $R$ is the removal pathway (i.e., sinks), and $\tau$ is atmospheric lifetime.**

| | $NH_3$ | $NH_4^+$ | $HNO_3$ | $NO_3^-\_f$ | $NO_3^-\_c$ | $SO_4^{2-}$ |
|---|---|---|---|---|---|---|
| Sources (TgN/TgS yr$^{-1}$) | 53.1 | 21.0 | 49.5 | 4.54 | 27.2 | 27.6 |
| Sinks (TgN/TgS yr$^{-1}$) | 55.6 | 20.8 | 49.4 | 4.52 | 26.9 | 27.2 |
| Dry deposition | 22.4 | 5.02 | 8.35 | 1.66 | 7.05 | 5.24 |
| Wet deposition | 12.2 | 15.8 | 9.25 | 2.87 | 19.8 | 22.0 |
| Chemical loss | 21.0 | - | 31.8 | - | - | - |
| Flux (out - in) | 0.00 | 0.01 | 0.00 | -0.01 | 0.02 | 0.00 |
| Burden (TgN/TgS) | 0.24 | 0.51 | 0.15 | 0.06 | 0.52 | 0.63 |
| Lifetime (days): | $\dfrac{Q(NH_3)}{P(NH_3)}$ | $\dfrac{Q(NH_4^+)}{P(NH_4^+)}$ | $\dfrac{Q(HNO_3)}{P(HNO_3)}$ | $\dfrac{Q(NO_3^-\_f)}{P(NO_3^-\_f)}$ | $\dfrac{Q(NO_3^-\_c)}{P(NO_3^-\_c)}$ | $\dfrac{Q(SO_4^{2-})}{P(SO_4^{2-})}$ |

| | $\tau = \dfrac{Burden}{Sources}$ | 1.6 | 8.9 | 1.1 | 4.8 | 7.0 | 8.3 |
|---|---|---|---|---|---|---|---|
| Lifetime (days): $\tau = \dfrac{Burden}{Sinks}$ | | $\dfrac{Q(NH_3)}{R(NH_3)}$ | $\dfrac{Q(NH_4^+)}{R(NH_4^+)}$ | $\dfrac{Q(HNO_3)}{R(HNO_3)}$ | $\dfrac{Q(NO_3^-\_f)}{R(NO_3^-\_f)}$ | $\dfrac{Q(NO_3^-\_c)}{R(NO_3^-\_c)}$ | $\dfrac{Q(SO_4^{2-})}{R(SO_4^{2-})}$ |
| | | 1.6 | 8.9 | 1.1 | 4.8 | 7.1 | 8.5 |

Differences in species lifetimes underpin the differences in vertical and horizontal distributions of the species. For example, as discussed in Sect. 3.1.1, gaseous $NH_3$, with a short lifetime, only shows dominant contribution to total RDN surface concentration in places near emission sources, whereas aerosol-phase $NH_4^+$, as a RDN reservoir species with a longer lifetime, is the major form of RDN over the remote continental regions and oceans.

The vertical distributions of $N_r$ and $S_r$ species shown in Fig. 10 also reflect the different species lifetimes. All species exhibit strong vertical gradients of decreasing concentration from the surface up to 10 km. The gaseous species ($NH_3$, $NO_2$ and $SO_2$) have peak concentrations at the surface between 20-40°N as shown in their surface distribution maps (Sect. 3.1.1). These gaseous species are emitted at the surface and their lifetimes are too short for significant mixing to high altitudes. Between the model surface layer (~50 m) and ~2.5 km the maximum zonal mean of $NH_3$ and $SO_2$ decreases from 1.77 µg m$^{-3}$ to 0.31 µg m$^{-3}$, and from 1.78 µg m$^{-3}$ to 0.23 µg m$^{-3}$, respectively, whereas the concentration gradient of $NO_2$ is steeper, with maximum zonal mean decreasing from 4.11 µg m$^{-3}$ to 0.10 µg m$^{-3}$, reflecting that production and removal processes of $NO_2$ mainly occur in the surface layer.

In contrast, the secondary species $NH_4^+$, $NO_3^-$ (fine + coarse) and $SO_4^{2-}$ are mixed to higher altitudes than their precursors. The zonal mean concentrations of the secondary species also show a larger horizontal variation range (10-60°N), again reflecting their longer transport times. A notable observation from Fig. 10 is that the maximum concentrations of the secondary aerosol species are not at the surface, a feature that will not be apparent from surface measurements of these species. The maximum zonal mean of $NH_4^+$ (1.20 µg m$^{-3}$) and $SO_4^{2-}$ (2.58 µg m$^{-3}$) occurs in the model level 88-177 m above the surface, whilst maximum $NO_3^-$ (3.80 µg m$^{-3}$) occurs in the 177-313 m model level. These elevated maxima in secondary components reflect their lack of surface emission and the time taken for the in-atmosphere oxidation and gas-to-particle chemistry.

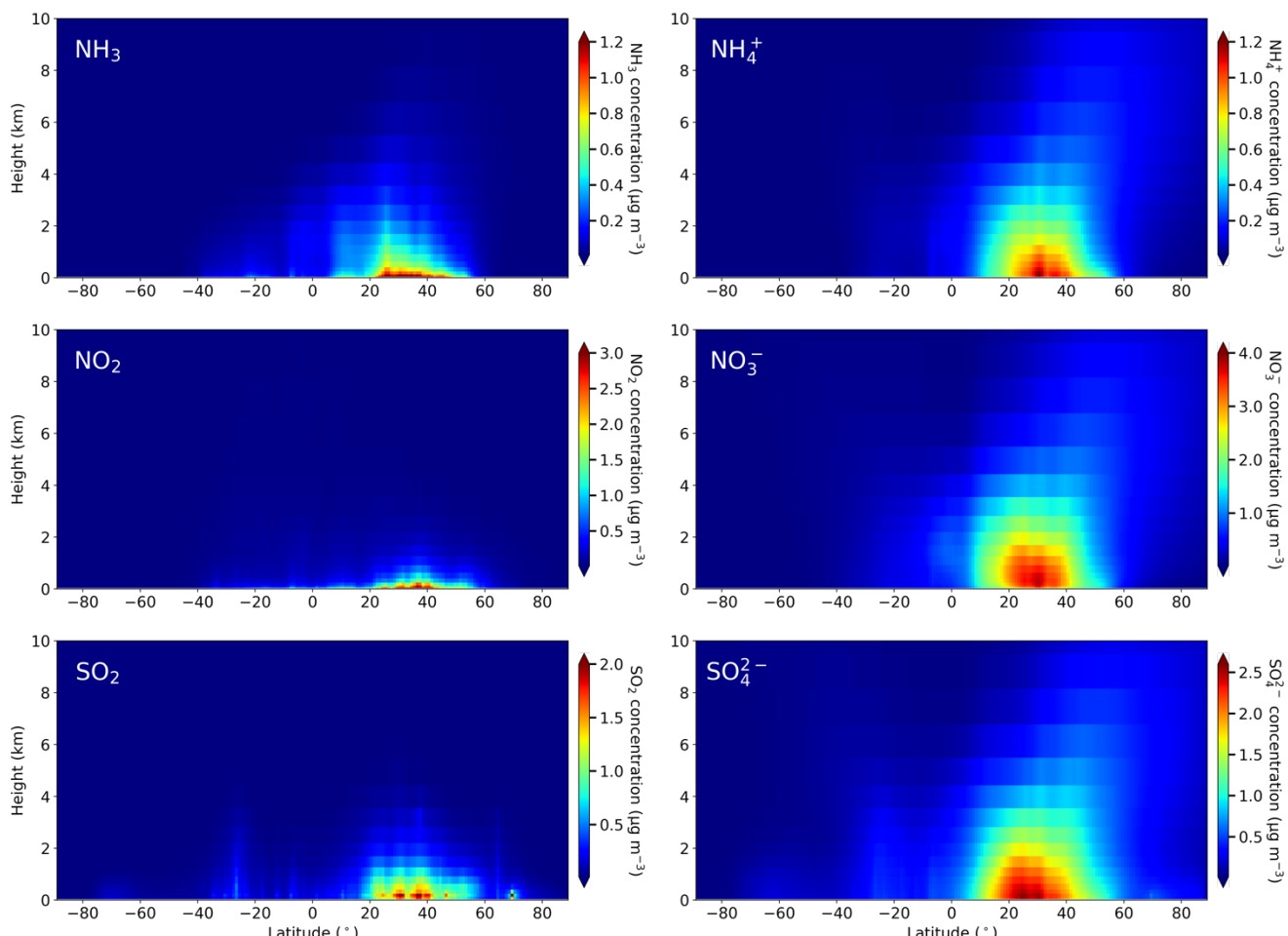

**Figure 10: Annual zonal mean vertical concentrations of NH$_3$, NH$_4^+$, NO$_2$, NO$_3^-$ (fine + coarse), SO$_2$ and SO$_4^{2-}$.**

### 3.3.3 Regional budgets

Figure 11 shows the differences between annual emissions and depositions for RDN, OXN, and OXS in the 10 world regions illustrated in Fig. 1. The corresponding absolute emission and deposition fluxes for each region are presented in Fig. S7.

For continental regions, the maximum fluxes for RDN are in East Asia (12.5 TgN yr$^{-1}$ for RDN emission and 12.0 TgN yr$^{-1}$ for RDN deposition), followed by South Asia (7.67 TgN yr$^{-1}$ emission and 5.46 TgN yr$^{-1}$ deposition) (Fig. S7). Both these regions are characterised by intensive agricultural activities and have RDN fluxes that are larger than the OXN fluxes. Fluxes of RDN in North America, Euro_Medi and Southeast Asia regions are lower (3.46-5.14 TgN yr$^{-1}$). The spatial distribution of RDN concentrations presented in Fig. 2 are the consequence of these regional variations in RDN fluxes. Figure 11 shows that

all regions are net exporters of RDN except for Southeast Asia and Rest of world in which marine areas dominate. The net import of RDN in Southeast Asis is, however, very small. In contrast, deposition of RDN in the Rest of world region is nearly 16 times its annual emissions. The difference of 7.89 TgN yr$^{-1}$ shows that RDN deposition is a notable source of continental anthropogenic pollution to the oceans. The region with greatest export of RDN is South Asia, at 2.21 TgN yr$^{-1}$ or 29% of its annual emissions (Fig. 11). Euro_Medi and Africa have the next greatest net exports, of 1.08 TgN yr$^{-1}$ (21% of emissions) and

0.95 TgN yr$^{-1}$ (15% of emissions), respectively. The relative differences between emissions and depositions over North America, South America and Oceania are in the range 6.0- 8.0%. Although East Asia is the region with greatest RDN emissions, it only exports 3.8% (0.48 TgN yr$^{-1}$). This region is geographically generally self-contained from the other regions

but means that East Asia is largely responsible for its own RDN pollution. The relatively high availability of acidic species also leads to shorter chemical lifetimes in this region.

For OXN, East Asia and Africa are the two continental regions with the largest annual emission and deposition fluxes (6.72-9.59 TgN yr$^{-1}$), followed by South America and North America (5.49-7.02 TgN yr$^{-1}$) (Fig. S7). In all of Southeast Asia, Euro_Medi, North America, South America, Africa, Oceania, and Rest of world the OXN fluxes have larger influence on their total nitrogen budgets than the RDN fluxes, whereas, as noted above, in East Asia and South Asia fluxes of RDN comprise the majority. In contrast to our finding of higher RDN flux in East Asia (12.0-12.5 TgN yr$^{-1}$) than OXN flux (8.33-9.59 TgN yr$^{-1}$), Tan et al. (2018) report higher OXN fluxes (10.5-11.4 TgN yr$^{-1}$) than RDN fluxes (7.7-8.5 TgN yr$^{-1}$). Such difference in RDN and OXN fluxes is partially attributed to differences in emission inventories, model structures, meteorology input, and region definition between the two studies. It may also reflect an increase in NH$_3$ emissions and a reduction in NO$_x$ emissions in this region between 2015 and 2010.

Figure 11 shows that the continental region with the greatest net export of OXN (1.92 TgN yr$^{-1}$, 22% of emissions) is Africa, followed by Euro_Medi (1.61 TgN yr$^{-1}$, 26%), South Asia (1.57 TgN yr$^{-1}$, 34%), and South America (1.51 TgN yr$^{-1}$, 22%). In particular, OXN deposition over Rest of world (13.4 TgN yr$^{-1}$) is around 5 times larger than its emission (2.21 TgN yr$^{-1}$, from shipping). The difference of 11.2 TgN yr$^{-1}$ between OXN deposition and emission in this region is greater than the 7.89 TgN yr$^{-1}$ difference for RDN. It is likely that the oxidation of NO$_x$ to reservoir organic nitrate species such as PAN facilitates the long-distance transport of OXN from the continent to the ocean, whereas the short lifetime of NH$_3$ means less RDN is transported from its continental emissions regions. Regardless of the explanation, the very large net deposition of OXN in the Rest of world region again indicates the scale of anthropogenic N deposition to marine areas.

Considering OXS, Fig. S7 shows that the Rest of world region has both the largest emission (10.6 TgS yr$^{-1}$) and deposition (17.4 TgS yr$^{-1}$) fluxes. The former is due to the substantial marine DMS emissions whose atmospheric oxidation ultimately produces SO$_2$ and H$_2$SO$_4$ (Marandino et al., 2013; Schobesberger et al., 2013; Hoffmann et al., 2016). These OXS emissions largely deposit back to the oceans, but Fig. 11 shows that OXS deposition in the Rest of world is also substantially augmented by long-range transport of OXS pollution from other regions. For continental regions, East Asia has the largest OXS fluxes, with emission and deposition reaching 8.95 TgS yr$^{-1}$ and 8.30 TgS yr$^{-1}$, respectively, followed by Rest of Asia (5.48-6.21 TgS yr$^{-1}$) and Euro_Medi (4.05-5.57 TgS yr$^{-1}$) (Fig. S7). South Asia is the largest net exporter of OXS with 1.62 TgS yr$^{-1}$ (37% of its emission) being deposited outside this region (Fig. 11). Euro_Medi is the second largest net exporter of OXS (1.52 TgS yr$^{-1}$, 27%). All other continental regions are also either positive or negligible exporters of OXS except for Southeast Asia which receives more OXS deposition than it emits. This is similar to the situation for RDN because away from major NO$_x$ emission areas NH$_4^+$ is predominantly associated with SO$_4^{2-}$.

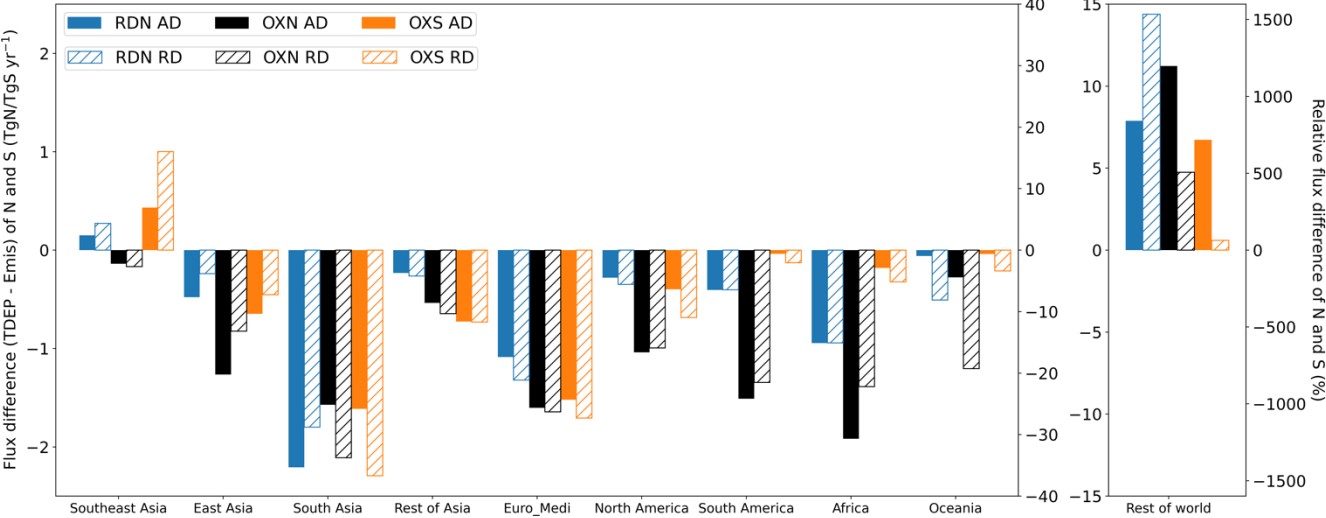

 **Figure 11: Absolute (AD) and relative differences (RD) between emission and deposition fluxes of RDN (TgN yr$^{-1}$), OXN (TgN yr$^{-1}$), and OXS (TgS yr$^{-1}$) for the 10 world regions defined in Fig. 1. Relative differences are expressed relative to emission fluxes. Difference values for the 'Rest of world' region are much larger than for other regions. The constituents of RDN, OXN and OXS are listed in Table S1. The corresponding emission and deposition fluxes are shown in Fig. S7.**

The tropospheric burden of each species is around one or two orders of magnitude smaller than its corresponding emission/deposition flux. Figure 12 and Fig. S8 respectively show, for each of the 10 world regions, the relative and absolute contributions of individual RDN and OXN species to the total tropospheric N burden, and the contributions of the individual OXS species to the total tropospheric S burden. In Table 3, we present the regional atmospheric lifetimes of RDN, OXN and OXS as quantified against their total emission or deposition fluxes. These regional lifetimes are not fully accurate because they do not take into account the net transport of RDN, OXN or OXS across the boundary of the region. However, given that, aside from the Rest of world region, the net transport in/out of a region is relatively small compared to the emission and deposition fluxes (Fig. 11), they should be reasonable estimates of the regional RDN, OXN and OXS lifetimes. Table 3 shows that the regional RDN, OXN and OXS lifetimes vary by up to a factor of 4 across the continental regions, being generally longest over Africa and Rest of Asia, where population and emissions are generally less dense, and shortest over East Asia, Southeast Asia and South Asia.

The Rest of world region has the highest burdens of all regions with 0.23 TgN RDN, 0.44 TgN OXN, and 0.27 TgS OXS, representing 31%, 37%, and 35% of global RDN, OXN and OXS burdens respectively, despite the relatively small emissions for RDN and OXN from this region. This observation is consistent with the conclusions from analyses of regional emission/deposition fluxes in Fig. 11 that the ocean areas are substantially impacted by transport of $N_r$ and $S_r$ air pollutants from continental regions. The lifetimes of RDN, OXN, and OXS in Rest of world are also the longest of all regions (Table 3), reflecting that RDN, OXN, and OXS in this region predominantly exist as longer-lived species: $SO_4^{2-}$ in the case of $S_r$, and $NH_4^+$, $NO_3^-$ and 'Rest OXN' (predominantly PAN) in the case of $N_r$. East Asia, Rest of Asia, and Africa have moderate burdens of 0.09-0.11 TgN (12%-14% of global burden) for RDN, 0.10-0.16 TgN (9%-13%) for OXN, and 0.08-0.12 TgS (11%-16%) for OXS. South Asia, Euro_Medi, and North America have smaller burdens with their contributions to global burdens ranging from 6%-9% for RDN, 4%-7% for OXN, and 5%-8% for OXS.

The species contributions to N burdens differ between the regions (Fig. 12). RDN contributes 58% and 49% to total N burden in South Asia and East Asia, respectively, which is similarly reflected in the emission and deposition fluxes in the two regions, followed by Rest of Asia (42%) and Euro_Medi (39%). The largest relative contributions of OXN to total N burden are in Oceania (82%) and South America (70%).

The largest contribution of $NH_3$ to RDN burden is in South America (67%) and this large proportion of $NH_3$ gives South America the shortest RDN lifetime (1.8-1.9 days) of any region. In other parts of the world, $NH_4^+$ is the majority contributor to RDN burden. The RDN lifetime in Southeast Asia and East Asia is relatively short (2.6-3.0 days), while Rest of Asia has the longest RDN lifetime (7.0-7.3 days) except for Rest of World.

For OXN, the contribution of $HNO_3$ to the OXN burden reaches 28%, 24% and 22% in Southeast Asia, Oceania, and South America, respectively, whereas $HNO_3$ contributes only 5%-12% to OXN in other parts of the world (Fig. 12). $NO_3^-$_f only makes a significant contribution (20%) to total OXN burden over East Asia; its contribution is <6% in other regions. This reflects the strong propensity for $NH_4NO_3$ formation in East Asia, which is similarly shown in the composition pattern of OXN deposition in this region as well (Sect. 3.2.2). Coarse $NO_3^-$ is a significant contributor to OXN burden, particularly in South Asia (59%), Rest of Asia (54%), and Africa (52%). This is associated with an abundance of coarse particles like sea salt, dust, and sand in these regions. It is therefore critical to correctly account for contributions of coarse nitrate in models as it overwhelmingly controls atmospheric nitrate burden and determines the size distribution of nitrate aerosol. In all world regions $SO_4^{2-}$ is the predominant form of the OXN burden (Fig. 12).

To mitigate atmospheric $N_r$ and $S_r$ burdens and their associated impacts on human health and ecosystems, tackling secondary inorganic species like $NH_4^+$, and coarse $NO_3^-$ and their precursors should be prioritised. The impacts of possible emission controls on atmospheric $N_r$ and $S_r$ concentrations, depositions, budgets, and lifetimes can be quantified by the emission sensitivity tests. Results from these experiments will be discussed in separate publication.

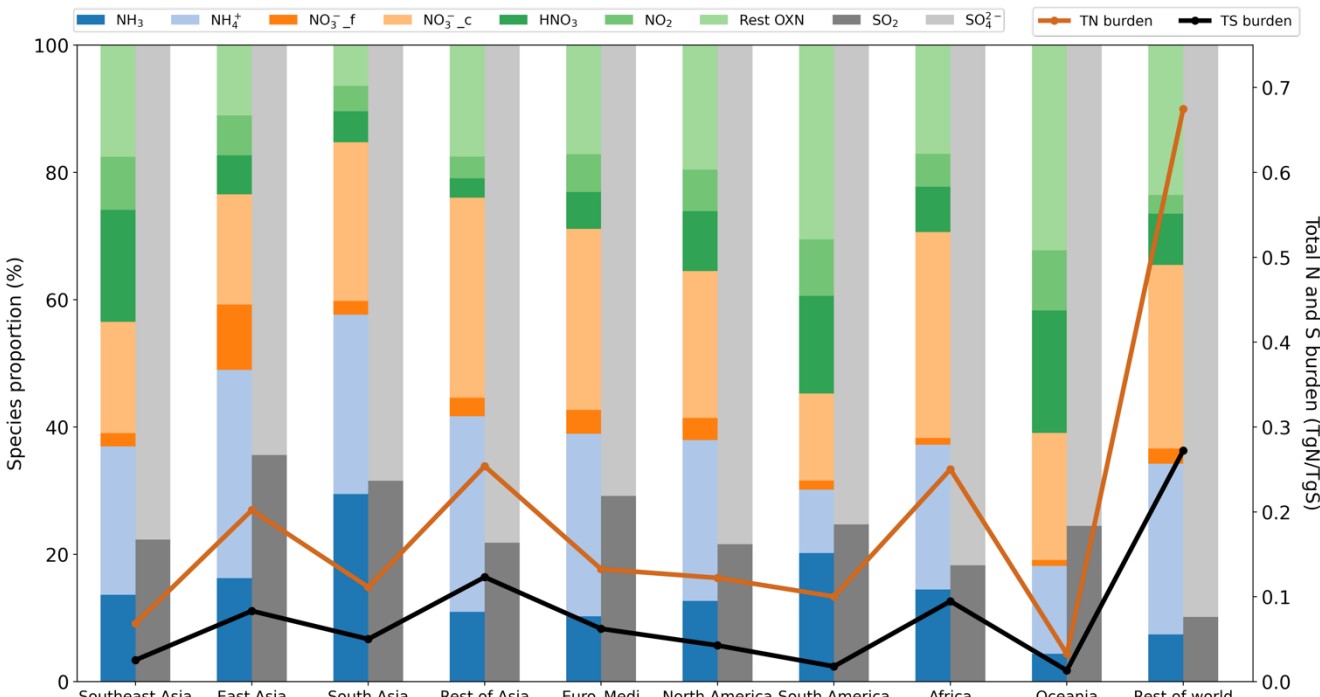

**Figure 12: Relative compositions of the tropospheric burdens in 2015 (stacked bar plots, left *y* axis) of total N (RDN + OXN) and total S (OXS) for the 10 world regions defined in Fig. 1. The total N (RDN + OXN) and S regional burdens are shown as the line plots (right *y* axis). 'Rest OXN' includes HONO, $N_2O_5$ and other OXN species (full listing given in Table S1).**

**Table 3: Regional atmospheric lifetimes for RDN, OXN and OXS calculated using either total emission or deposition fluxes. *Q* is the burden, *P* is the production pathway (i.e., emissions), *R* is the removal pathway (i.e., depositions), and *τ* is atmospheric lifetime. These regional lifetimes are not fully accurate because they do not take into account the net transport of RDN, OXN or OXS across the boundary of the region, but these are relatively small for most continental regions. Please note that the lifetimes derived from the production pathway for Rest of world are not listed in this table because these numbers are highly skewed by the the net transport of RDN, OXN and OXS from other regions (full data is available in Table S5).**

| (days) | Southeast Asia | East Asia | South Asia | Rest of Asia | Euro_Medi | North America | South America | Africa | Oceania | Rest of world |
|---|---|---|---|---|---|---|---|---|---|---|
| | | | | | RDN | | | | | |
| $\tau = \frac{Q}{P}$ | 2.7 | 2.9 | 3.1 | 7.0 | 3.7 | 3.4 | 1.8 | 5.4 | 2.9 | - |
| $\tau = \frac{Q}{R}$ | 2.6 | 3.0 | 4.3 | 7.3 | 4.6 | 3.6 | 1.9 | 6.4 | 3.2 | 10 |
| | | | | | OXN | | | | | |
| $\tau = \frac{Q}{P}$ | 3.1 | 3.9 | 3.7 | 10 | 4.8 | 4.2 | 3.6 | 6.6 | 6.8 | - |
| $\tau = \frac{Q}{R}$ | 3.1 | 4.5 | 5.6 | 12 | 6.6 | 5.0 | 4.6 | 8.5 | 8.4 | 12 |
| | | | | | OXS | | | | | |
| $\tau = \frac{Q}{P}$ | 3.4 | 3.4 | 4.2 | 7.2 | 4.1 | 4.3 | 3.6 | 9.9 | 4.2 | - |
| $\tau = \frac{Q}{R}$ | 3.0 | 3.7 | 6.6 | 8.2 | 5.6 | 4.8 | 3.7 | 10 | 4.4 | 5.7 |

The model evaluation presented in Ge et al. (2021) shows that the model-measurement bias varies from one pollutant to another. For instance, the model underestimates annual global $HNO_3$ surface concentration measurement by 34%, but overestimates $NO_2$ by 23%. Although it is difficult to diagnose the extent to which bias is contributed by bias intrinsic to the model, including in emissions input to the model, or to bias in measurements, or to measurement-model spatial representation mismatch, it is important to acknowledge that all numbers reported in this work may be biased somewhat low/high.

The intention here is to provide a new overview of how $N_r$ and $S_r$ budgets vary among different world regions and among different chemical species. Since Ge et al. (2021) show that the EMEP model is capable of capturing the overall spatial variation of surface concentrations for major $N_r$ and $S_r$ pollutants and their wet deposition, and since our results are comparable with similar global modelling studies, we have confidence that we characterise the major regional differences in budgets. However, we also acknowledge that there are varying uncertainties in different regions, as revealed by our model evaluation study (Ge et al., 2021). We therefore have greater confidence in regional budgets in North America, Europe, and Southeast Asia than other world regions. But whether different regional uncertainties derive from the emissions, model processes, measurements, or a combination of these aspects, is impossible to resolve only from a modelling study. Further evaluation studies of emissions and measurements are required to grain more insight.

## 4 Conclusions

We used the EMEP MSC-W model 4.34 coupled with WRF 4.2.2 meteorology to undertake a new present-day evaluation of global and regional concentrations, wet and dry deposition, fluxes and lifetimes of reactive N and S species. These are quantities that cannot be derived from measurements alone. Simulations used the 2015 ECLIPSE annual emissions with EDGAR monthly profiles. Surface concentration and wet deposition outputs from this model configuration have been comprehensively evaluated against several measurement networks.

Our simulations show that the largest surface concentrations and deposition of RDN ($NH_3 + NH_4^+$), OXN ($NO_x + HNO_3 + HONO + N_2O_5 + NO_3^-\_f + NO_3^-\_c$ + other OXN species), and OXS ($SO_2 + SO_4^{2-}$) are in eastern China and northern India. In these areas RDN concentrations are predominantly in the form of $NH_3$ ($NH_4^+$ is typically <20%), as also in Europe and North America. $NH_4^+$ comprises >90% of RDN over oceans and remote continental areas. Gaseous OXN species constitute the majority of OXN over continental regions, whereas $NO_3^-$ dominates OXN over oceans, except for major shipping routes. In eastern China, Europe, and eastern North America fine $NO_3^-$ dominates nitrate, but in India coarse $NO_3^-$ dominates, as has been noted elsewhere. Aerosol $SO_4^{2-}$ is the major form of OXS nearly everywhere except in the vicinity of very large $SO_2$ emission sources.

Most continental regions are now characterised as 'ammonia rich', which indicates that whilst reducing $NH_3$ emissions will decrease RDN concentration and deposition it will have little effect on mitigating secondary inorganic aerosol. Our results show an increase in ammonia richness compared with model simulations for 2000 in other studies. This is driven not only by increasing $NH_3$ emission in parts of the world, but also because a growing number of countries have been reducing $NO_x$ and $SO_x$ emissions much faster than $NH_3$, for which many countries have yet to set emissions targets. In general, South Asia shows the most ammonia-rich condition. Coastal areas around East Asia, northern Europe, and north-eastern United States can be characterised as 'nitrate rich' where $NH_4NO_3$ formation is limited by $NH_3$ not nitrate. These locations experience transport of oxidised N from the adjacent continent and/or direct shipping emissions of $NO_x$ but $NH_3$ concentrations are lower because it deposits more quickly to land before transport off the coast.

Our modelling shows that OXN (57.9 TgN yr$^{-1}$, 51%) and RDN (55.5 TgN yr$^{-1}$, 49%) account for almost equal proportions of the total nitrogen deposition. Total sulphur (OXS) deposition is 50.5 TgS yr$^{-1}$. Globally, wet and dry deposition contribute equally to RDN deposition. For OXN and OXS, wet deposition contributes slightly more than dry deposition. However, the contributions of different deposition components vary regionally. An advantage of modelling is that it can distinguish the

individual deposition processes, which measurements cannot. For example, the model shows that in many regions the majority of the wet deposition of RDN derives from rainout of gaseous $NH_3$ rather than of particle $NH_4^+$. For RDN deposition, dry deposition of $NH_3$ is the predominant contributor in most regions except for Rest of Asia and Rest of world (essentially oceans) where $NH_3$ emissions are small and RDN deposition is largely determined by transport of aerosol $NH_4^+$ from other regions. Further reductions in $NH_3$ would thus efficiently reduce the deposition of RDN in most world regions. The two largest contributors to OXN deposition in all regions are $HNO_3$ and coarse $NO_3^-$ (both occurring by both wet and dry deposition). The deposition of fine $NO_3^-$ is only important over East Asia (24%) and is <10% in other regions. For OXS deposition, dry deposited $SO_2$ and wet deposited $SO_4^{2-}$ are two largest contributors in all regions.

The annual average tropospheric burden of RDN is 0.75 TgN of which $NH_3$ and $NH_4^+$ comprise 32% (0.24 TgN) and 68% (0.51 TgN) respectively. The average tropospheric lifetime of $NH_3$ is 1.6 days, while that of $NH_4^+$ is more than 5 times longer (8.9 days). The annual average burden of $HNO_3$ is 0.15 TgN with an average lifetime of only 1.1 days. Total nitrate burden is 0.58 TgN with fine $NO_3^-$ only constituting 10% of this total. The lifetime of fine nitrate (4.8 days) is shorter than that of coarse nitrate (7.0-7.1 days). The global lifetime of OXN (7.6-7.7 days) is longer than that of RDN (4.9-5.2 days), consistent with a total OXN burden (1.20 TgN) almost double that of RDN. The tropospheric burden of OXS is 0.78 TgS with a global lifetime of 5.6-5.9 days. The $SO_4^{2-}$ burden is 0.63 TgS and its lifetime of 8.3-8.5 days is similar to that of $NH_4^+$, reflecting the close association between the two species in the atmosphere globally. It is notable that lifetimes of RDN, OXN, and OXS vary by a factor of 4 across different continental regions. East Asia has quite short lifetimes for RDN (2.9-3.0 days), OXN (3.9-4.5 days), and OXS (3.4-3.7 days), whereas the lifetimes in Rest of Asia and Africa are around twice as long, which is related to the regional differences in meteorological conditions and compositions of RDN, OXN, and OXS burdens.

The model confirms that the longer lifetimes of the secondary $N_r$ and $S_r$ species are consistent with their greater vertical transport, which surface measurements cannot reveal. In addition, whilst the gaseous precursor species have maximum concentrations at the surface, maximum concentrations of the aerosol species are at higher altitudes because of the time taken for their formation.

Comparison of the absolute emission and deposition fluxes of RDN and OXN in different world regionns show that East Asia and South Asia are characterised by larger RDN fluxes than OXN fluxes, while most other world regions exhibit larger OXN fluxes than RDN fluxes, which likely reflects greater increases in $NH_3$ emissions in the two regions.

Comparison of the difference between emission and deposition fluxes shows that South Asia region is the largest net exporter of RDN (2.21 TgN yr$^{-1}$, 29% of its annual emission), followed by Euro_Medi and Africa and other continental regions. Despite having the largest RDN emissions and depostion, East Asia has very small net export and is therefore largely responsible for its own RDN pollution. In contrast, Rest of world, which is mainly the ocean regions (as well as Southeast Asia which also contains a lot of marine area) receive more deposition of RDN than they emit.

For OXN, Africa is the largest net exporter of OXN at 22% (1.92 TgN yr$^{-1}$) of its emission, followed by South Asia. The Rest of world again receives substantial net import of OXN (11.2 TgN yr$^{-1}$) which is even higher than the 7.89 TgN yr$^{-1}$ net import of RDN. Oxidation of $NO_x$ to reservoir organic nitrate species such as PAN likely facilitates the long-distance transport of OXN from the continent to the ocean, whereas the short lifetime of $NH_3$ means less RDN is transported from its continental emissions regions. The large flux imbalances of OXN and RDN in these areas indicates the scale of anthropogenic N pollution to marine areas.

The two regions with the largest net export of OXS are South Asia (1.62 TgS yr$^{-1}$, equivalent to 37% of its emission) and Euro_Medi (1.52 TgS yr$^{-1}$, 27%). All other continental regions are either positive or zero exporters of OXS except for Southeast Asia which receives more OXS depostion than it emits. This is similar to the situation for RDN because away from major $NO_x$ emission areas $NH_4^+$ is predominantly associated with $SO_4^{2-}$.

The three largest contributors to total N burden globally are $NH_4^+$, coarse $NO_3^-$, and 'Rest OXN' (mainly PAN). In South Asia, Rest of Asia, and Africa, coarse $NO_3^-$ constitutes over half of OXN burden, attributable to an abundance of coarse sea-salt and dust particles in these regions. It is critical to account for contributions of coarse nitrate as it controls atmospheric nitrate burden and the size distribution of nitrate aerosol. Aerosol $SO_4^{2-}$ is the predominant form of OXS in all regions.

As presented in this work, $N_r$ and $S_r$ species are abundant air pollutants impacting gas and particle concentrations and deposition globally. The chemical and meteorological linkages between emissions and $N_r$ and $S_r$ species concentrations are complex, and the detail can only be revealed by process-based atmospheric modelling as illustrated here, despite the uncertainties in emissions and model parameterisations.

**Code and data availability**

As described and referenced in Section 2 of this paper, this study used two open-source global models: the European Monitoring and Evaluation Programme Meteorological Synthesizing Centre -West atmospheric chemistry transport model (EMEP MSC-W, version 4.34, source code available at https://doi.org/10.5281/zenodo.3647990) and the Weather Research and Forecast meteorological model (WRF, version 4.2.2, www.wrf-model.org, https://github.com/wrf-model/WRF/releases/tag/v4.2.2). The model output presented in figures and tables in this paper and the corresponding Python scripts are available at https://doi.org/10.5281/zenodo.5912055.

**Author contribution**

MH, DS and MV conceptualised and supervised the study. MV and PW contributed to model development and set-up and provided modelling support. MV provided computing resource. YG contributed to study design, undertook all model simulations, formal data analyses, visualisation of the results and data curation, with discussion and refinement by all authors. The original draft of the paper was written by YG with contributions and editing by MH. All authors provided review comments and approval of the final version.

**Competing interests**

The authors declare that they have no conflict of interest.

**Acknowledgments**

Y. Ge gratefully acknowledges studentship funding from the University of Edinburgh and its School of Chemistry. This work was in part supported by the UK Natural Environment Research Council (NERC), including grant nos. NE/R016429/1 and NE/R000131/1 as part of the UK-SCAPE and SUNRISE programmes delivering UK National Capability, and the European Modelling and Evaluation Programme under the United Nations Economic Commission for Europe Convention on Long-range Transboundary Air Pollution.

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
