# Peer review of "A new assessment of global and regional budgets, fluxes and lifetimes of atmospheric reactive N and S gases and aerosols"

_Atmospheric Chemistry and Physics, 2022_

## Referee Comment (RC1)

**Review of Ge et al. ACP-2022-82**

**General comments:**

The manuscript provides a comprehensive analysis of the concentrations, wet and dry deposition, fluxes, and lifetimes of reactive N and S gases and aerosols globally, and for 10 world regions. The goal of the manuscript is to conduct a more up-to-date examination of atmospheric processes affecting the fate of reactive N and S, particularly of the reduced N species  $NH_3$  and  $NH_4^+$ , for developing mitigation strategies according to different regional conditions. Overall, this study is an important contribution to the field as it highlights the substantial regional variation in  $N_r$  and  $S_r$  budgets and the need for modelling to simulate the chemical and meteorological linkages underpinning atmospheric responses to precursor emissions. The manuscript is well written and structured clearly and fits well the scope of this journal. I recommend that this study can be accepted for publication after the specific comments below are addressed.

**Major comments:**

- 1. Section 3.1.2, lines 225-228: What was the rationale for choosing the fine SIA concentration of 0.5  $\mu$ g m-3 as the threshold?
- 2. Section 3.3.1, lines 460-464: To increase study reproducibility and transparency, the authors may want to consider providing detailed calculation equations of atmospheric burdens and lifetimes (perhaps adding a section in the supplementary).
- 3. Section 3.3.1, lines 517-520: In general, gaseous NH3 deposits quicker than aerosolphase NH4+. Could you provide more information on the difference between NH3 deposition and NH4+ deposition in the model as deposition is crucial for lifetime calculations?
- 4. Section 3.3.3, lines 618-619: Is there any explanation for why does South America have such large proportion of RDN in the form of NH3?

**Technical corrections:**

Line 758: DOI information is missing in this citation. Please add the "doi" number before "2013b".

Line 809: Please add the "doi" number before "2005".

Line 896: Please add the "doi" number before "2015". Line 966: Please add the "doi" number before "2012".

---

## Author Comment (AC1)

**acp-2022-82: A new assessment of global and regional budgets, fluxes and lifetimes of atmospheric reactive N and S gases and aerosols**
**Ge et al.**

5 **Response to Reviewer #1**

We thank the reviewer for their time spent reading our manuscript. Below we include all the reviewer comments and provide in blue text our point-by-point responses. All line numbers mentioned in our responses refer to the clean revised manuscript (not the track-changed version).

**General comments:**

This paper presents detailed analysis of nitrogen and sulfur fluxes using a global model run from 2015. The results give an updated picture of chemical processing and interregional transport and deposition

15 of these key pollutants, and important recommendations for effective mitigation efforts aimed to improve air quality. The manuscript is well-written and organized, and I recommend publication following minor revisions discussed below.

Response: We thank the reviewer for their support of our work and for their recommendation of

20 publication after attention to some minor revisions.

**Specific comments:**

**Specific comments** 1. Please include some discussion of model performance and uncertainty.

25

Response: As the reviewer mentions, detailed discussions about the model performance for simulating concentrations and wet deposition of $N_r$ and $S_r$ species are presented in Ge et al. (2021), which we cited in our paper. For our revised paper we have now added the following summary of findings from Ge et al. (2021) in a new Sect. 2.3 'Model performance' (lines 144-173).

30

"Ge et al. (2021) provide a comprehensive evaluation of surface concentrations and wet deposition of $N_r$ and $S_r$ species from this model configuration against global measurements from 10 monitoring networks. Comparisons of 2010 and 2015 surface concentrations between model and measurement demonstrate that the model captures well the overall spatial and seasonal variations of the major

35 inorganic pollutants $NH_3$, $NO_2$, $SO_2$, $HNO_3$, $NH_4^+$, $NO_3^-$, $SO_4^{2-}$, and their wet deposition in East Asia, Southeast Asia, Europe, and North America. Correlation coefficients between modelled and measured annual mean concentrations of all species are >0.78 except for $HNO_3$ and $SO_2$, and are 0.78 and 0.63

for annual wet deposition of reduced N and oxidised N, respectively. Given the intrinsic scale mismatch between a local site measurement and a global-scale chemistry model grid, these comparisons are good, and are comparable with model evaluation statistics determined for models of similar resolution (Hauglustaine et al., 2014; Bellouin et al., 2011; Pringle et al., 2010; Xu and Penner, 2012).

Both model and measurement have uncertainty that constrains the extent to which statistical analyses between modelled and measured data can be utilized to assess a model's performance. Model uncertainty may come from uncertainty in emission, meteorology inputs, and computing processors, and from parameterisations of chemical/physical processes, while measurement uncertainty comes from sampling and analytical procedures. Emission inventory compilation is partially based on reported measurement data and partially on expert estimation, which complicates assigning uncertainties to both emission magnitudes and their spatiotemporal profiles. It is therefore not possible to quantify the contribution of emissions uncertainty to model uncertainty, but it will surely vary regionally and for different species. We have shown in Ge et al. (2021) that modelled concentrations of secondary pollutants are less sensitive to the choice of emission inventory than for primary pollutants since the former are influenced by multiple emissions and the timescales for their formation act to smooth out the differentials in primary emissions. On the other hand, if it is variation in measurement accuracy that contributes to regional variation in model-measurement agreement, then confidence in model output is maintained. Ge et al. (2021) showed better evaluation statistics with measurements in Southeast Asia, Europe, and North America than in East Asia, which suggests shortcomings in regional measurements and/or emissions in the latter network rather than substantial systematic issues with model parameterisations of chemical/physical processes. Ge et al. (2021) also evaluated model response to changes between 2010 and 2015 ECLIPSE$_E$ emissions and showed that modelled concentration changes of primary and secondary pollutants and of deposition of reduced N, oxidised N, and oxidised S, were consistent with the trends in the corresponding measurements, given realistic uncertainties in emissions and measurements and of meteorological influences between the two years.

Nevertheless, considering the fundamental uncertainties in emission, model and measurements, all numbers reported in this work should be considered as having underlying uncertainty, albeit that the latest available emission inventory and model versions were used to minimize as far as possible uncertainties from the first two of these.

**Specific comments** 2. How would the results of the measurement evaluation in the preceding model evaluation paper (Ge et al., 2021, https://doi.org/10.5194/gmd-14-7021-2021) impact the main conclusions in this one? For example, HNO$_3$ concentrations were generally biased low compared to measurements. If this is due to its chemical production being too slow in the model, the resulting

deposition estimates would also be biased low; but if the concentration bias is due to deposition being

75      too fast, the resulting deposition may be biased high.

A fundamental difference between model and measurement must first be acknowledged. Even if both model and measurements were perfect representations, there still would not be complete agreement because a measurement is for a single point in space at a specific height above the ground, whose

80      concentration may well not reflect the average concentration within our global model simulation surface layer grid of a volume of $1° \times 1° \times 50$ m (around 500 km$^3$). The existence of bias between model and measurement is therefore inevitable, but since the magnitude of this source of bias is not known, it hinders determination and quantification of biases intrinsic to the model itself. In addition, model-measurement bias may also arise because of issues with the measurement value as well or in instead of

85      issues intrinsic to the model. $HNO_3$ is harder to measure as accurately as some other species. As the reviewer notes, the $HNO_3$ concentrations were biased low in the model, but for the reasons described above it is not readily possible to diagnose the exact source of the bias: the reviewer's two model process suggestions are possible, but so are issues with measurement (e.g., intercomparison campaigns using different experimental methods for measuring gaseous $HNO_3$ have found relatively poor agreement

90      between different techniques (Hanke et al., 2003)) and with measurement-model spatial mismatch. If there is an issue within the model, then the largest impact of model underestimation may be on the calculation of global $HNO_3$ lifetime. A slower production pathway results in a longer lifetime. In contrast, a faster removal pathway results in a shorter lifetime. To acknowledge the reviewer's comment, we have added the following text (lines 691-695) in our revised paper:

95

"The model evaluation presented in Ge et al. (2021) shows that the model-measurement bias varies from one pollutant to another. For instance, the model underestimates annual global $HNO_3$ surface concentration measurement by 34%, but overestimates $NO_2$ by 23%. Although it is difficult to diagnose the extent to which bias is contributed by bias intrinsic to the model, including in emissions input to the

100     model, or to bias in measurements, or to measurement-model spatial representation mismatch, it is important to acknowledge that all numbers reported in this work may be biased somewhat low or high."

**Specific comments** 3. Additionally, were there regions where the model performed better/worse than others, and by extension where you have more/less confidence in the regional budgets presented here?

105

Again the answer is not straightforward. The quality of model-measurement comparison is determined not only by the 'internal' quality of the model but also by factors 'external' to the model such as the quality of relevant spatiotemporal emissions and the quality of the measurement. When a global model shows poorer agreement with measurement in some regions than others then it suggests that the issue

110     is more associated with discrepancies between different measurement networks and emissions reported

by different regions/countries than with the model. Insight into levels of regional confidence is also limited by the generally limited spatial coverage of a measurement network, even where there is such a network, and the lack of any extended measurements in many parts of the world. From a statistical point of view, a small number of localised disagreements between model and measurement may not be representative enough to draw a conclusion. It therefore does not follow that we have different confidence in our model output in different regions purely based on differences in model comparisons with surface measurements. Having said that, however, as we write in our new Section 2.3 on model performance uncertainty in emissions inventories is likely greater in some regions than others. Therefore, in addition to our text on regional differences within our new Section 2.3 we have now added the following text (lines 696-704) to the end of our Results and Discussion section:

"The intention here is to provide a new overview of how $N_r$ and $S_r$ budgets vary among different world regions and among different chemical species. Since Ge et al. (2021) show that the EMEP model is capable of capturing the overall spatial variation of surface concentrations for major $N_r$ and $S_r$ pollutants and their wet deposition, and since our results are comparable with similar global modelling studies, we have confidence that we characterise the major regional differences in budgets. However, we also acknowledge that there are varying uncertainties in different regions, as revealed by our model evaluation study (Ge et al., 2021). We therefore have greater confidence in regional budgets in North America, Europe, and Southeast Asia than other world regions. But whether different regional uncertainties derive from the emissions, model processes, measurements, or a combination of these aspects, is impossible to resolve only from a modelling study. Further evaluation studies of emissions and measurements are required to gain more insight."

**Specific comments** 4. Finally, while dry deposition was not evaluated against measurements (given the dearth of those), some caveats about the variability of dry deposition estimates would be appropriate.

As response to this comment we have added the following caveat (lines 327-329) to our revised paper:

"Additionally, it must be stressed that the lack of dry deposition comparison between model and measurement, due to the lack of measurement data, means that the numbers reported for modelled dry deposition in this section should be regarded as more uncertain."

L422-425: A 40% decrease in only 5 years seems unrealistic. Are the emission inventories comparable?

Response: The total S deposition (84 TgS yr$^{-1}$) estimated by Tan et al. (2018) is a multi-model mean of 11 global models using the HTAP global emission inventory for 2010, while our model simulation uses the ECLIPSE emission inventory for 2015. Additionally, different choices of natural emissions of S

(e.g., oceanic DMS, volcano emissions) are likely to influence the modelled total S deposition as well. In our previous model experiments we used both HTAP (2010) and ECLIPSE (2010 and 2015) emission inventories. Our modelled global OXS deposition based on different emission inventories is listed in Responses Table R1. First, our modelled total OXS deposition using HTAP 2010 emission inventory is 69.1 TgS yr$^{-1}$, which is still 18% smaller than the 84 TgS yr$^{-1}$ reported by Tan et al. (2018) despite using the same inventory and the same meteorology year. This likely reflects differences in the natural emissions of S between Tan et al. (2018) and our study. Secondly, in our study, under the same model configuration, natural emission input, and meteorology input, the difference of OXS deposition between HTAP 2010 and ECLIPSE 2010 is 9.3 TgS (13% and 15% with respect to HTAP and ECLIPSE respectively). This reflects the difference between the two emission inventories. Thirdly, in our study, using ECLIPSE 2010 and 2015 inventories, the total OXS deposition in 2015 is 16% (9.3 TgS) smaller than that in 2010, which likely reflects the reduction in global S emission from 2010 to 2015. Considering all these observations, the deposition difference between our work and Tan et al. (2018) is a combination of differences in inventory compilations, natural emission input, and actual S emission reductions from 2010 to 2015. We therefore concur with the reviewer that our original statement that differences were due to emissions reductions was too strong. To reflect this, we have rephrased our previous statement (lines 455-458) to the following text in our revised paper:

"The lower OXS deposition value presented here is assumed in part to be due to the considerable reduction in global sulfur emission from 2000 to 2015 (Fowler et al., 2020; Aas et al., 2019). A direct comparison of modelled S deposition between our work and other studies is confounded by use of different inventories for both anthropogenic and natural emissions."

**Table R1. Global OXS emissions and deposition from our model simulations using HTAP emission inventory in 2010, and ECLIPSE emission inventories in 2010 and 2015.**

| Unit: TgS | HTAP 2010 | ECLIPSE 2010 | ECLIPSE 2015 |
|---|---|---|---|
| Total emission | 65.6 | 56.6 | 48.5 |
| Total deposition | 69.1 | 59.8 | 50.5 |

L486-511: Given that the aerosol scheme includes equilibrium of ammonium nitrate (R4 in section S1), how do you handle that reaction in the lifetime calculations for $HNO_3$, $NH_3$, $NH_4^+$ and $NO_3^-\_f$?

Response: Although the formation of ammonium nitrate is a reversible reaction, the annual production of $NH_4NO_3$ is larger than its decomposition, otherwise fine nitrate $NO_3^-\_f$ would not deposit. In other words, the annual net direction of this equilibrium is from $HNO_3$ and $NH_3$ to $NH_4NO_3$. As shown in Responses Fig. R1, for a global domain chemical production is the only source of $HNO_3$, and its sinks

include wet and dry deposition, chemical conversion to NO$_3^-$_f and NO$_3^-$_c, and the flux loss from the model domain to the upper atmosphere (details in next paragraph). The wet and dry deposition is then the only removal pathway for NO$_3^-$_f and NO$_3^-$_c apart from their own flux losses to the upper atmosphere (i.e., outside the model domain). If the chemical production rate of HNO$_3$ is larger than its total removal rate, the mass of HNO$_3$ will accumulate in the atmosphere, whereas if its removal rate is larger than the production rate, the mass of HNO$_3$ will decrease after a one-year simulation.

For each species in the model, there are outputs of initial mass ($M_{initial}$, at the beginning of the simulation), final mass ($M_{final}$, at the end of simulation), Flux out ($Flux_{out}$, from the top of the model domain), and Flux in ($Flux_{in}$, from the top of the model domain; usually negligible). The rate of mass change ($M_{final} - M_{initial}$) for HNO$_3$ thus represents the difference between sources and sinks of HNO$_3$, which can then be used to calculate the production rate of HNO$_3$. The flux losses and rate of mass change are generally one or two orders of magnitude smaller than the deposition and chemical production quantities. We use equations Eq1-Eq8 to estimate the production and removal rates (TgN yr$^{-1}$) of HNO$_3$, NO$_3^-$_f and NO$_3^-$_c. Assuming steady-state condition:

$$F_{out-in}(species) = Flux_{out}(species) - Flux_{in}(species) \qquad (Eq1)$$

$$\Delta M(species) = M_{final}(species) - M_{initial}(species) \qquad (Eq2)$$

$$R\left(NO_{3\,f}^-\right) = DEP\left(NO_{3\,f}^-\right) + F_{out-in}\left(NO_{3\,f}^-\right) \qquad (Eq3)$$

$$R\left(NO_{3\,c}^-\right) = DEP\left(NO_{3\,c}^-\right) + F_{out-in}\left(NO_{3\,c}^-\right) \qquad (Eq4)$$

$$P\left(NO_{3\,f}^-\right) = \Delta M\left(NO_{3\,f}^-\right) + R\left(NO_{3\,f}^-\right) \qquad (Eq5)$$

$$P\left(NO_{3\,c}^-\right) = \Delta M\left(NO_{3\,c}^-\right) + R\left(NO_{3\,c}^-\right) \qquad (Eq6)$$

$$R(HNO_3) = DEP(HNO_3) + P\left(NO_{3\,f}^-\right) + P\left(NO_{3\,c}^-\right) + F_{out-in}(HNO_3) \qquad (Eq7)$$

$$P(HNO_3) = \Delta M(HNO_3) + R(HNO_3) \qquad (Eq8)$$

where $F_{out-in}(species)$, $\Delta M(species)$, $R(species)$, $DEP(species)$, and $P(species)$ represent the flux loss, mass change, removal rate, total deposition, and production rate for each species respectively.

[Figure]

**Figure R1. Basic chemical and physical behaviour of HNO₃ in the atmosphere. CHEM1 and CHEM2 represent chemical production of coarse and fine nitrate respectively. DEP represents total deposition for the indicated species.**

For NH₃ and NH₄⁺, the calculation principle is the same. The removal of NH₃ consists of its own deposition ($DEP(NH_3)$), $Flux_{out-in}$, and chemical loss to NH₄⁺ (Fig. R2). Therefore, the NH₄⁺ production rate equals NH₃ chemical loss rate, while the removal pathway of NH₄⁺ includes its own deposition and flux loss. The calculations of sources and sinks for NH₃ and NH₄⁺ are listed in equations Eq9-Eq11. Assuming steady-state condition:

$$R(NH_4^+) = DEP(NH_4^+) + F_{out-in}(NH_4^+) \qquad (Eq9)$$
$$P(NH_4^+) = \Delta M(NH_4^+) + R(NH_4^+) \qquad (Eq10)$$
$$R(NH_3) = DEP(NH_3) + P(NH_4^+) + F_{out-in}(NH_3) \qquad (Eq11)$$

Once the production and removal rates are obtained, the lifetime calculations for HNO₃, NH₃, NH₄⁺, NO₃⁻_f and NO₃⁻_c, utilizing their respective global burden data, follow the equations in Table 2 in the main paper.

[Figure]

**Figure R2. Basic chemical and physical behaviour of NH₃ in the atmosphere. DEP represents total deposition. CHEM1 and CHEM2 represent chemical production of ammonium sulfate and ammonium nitrate respectively.**

For this revised paper, we have now added a new section in the Supplementary Information (Sect. S1.2, lines 100-145) that comprises the above detailed description of how we calculate the atmospheric burdens and lifetimes that we present in the main paper.

L500: It's surprising that coarse nitrate would have a longer lifetime than fine, given their relative wet scavenging efficiencies. Could you explain? Is it due to the regional differences in where NO₃⁻_f and NO₃⁻_c are dominant?

Response: The longer global mean lifetime of coarse nitrate (7.0 days) than of fine nitrate (4.8 days) is consistent with a larger $NO_3^-$_c burden (0.52 TgN) compared to $NO_3^-$_f (0.06 TgN) in 2015. The dominance of coarse nitrate in global total nitrate burden is also observed in other global modelling studies. Hauglustaine et al. (2014), Xu and Penner (2012), and Bauer et al. (2007) reported a relative contribution of coarse-mode nitrate of 72%, 77%, and 79% to total nitrate burden in their respective models for 2000. Additionally, the lifetime of fine nitrate calculated in our work is comparable with other literature values. Park et al. (2004) reported a global fine nitrate lifetime against deposition of 3.2 days for 2001, Feng et al. (2007) reported a global lifetime of total nitrate (fine + coarse) of 5.0 days for 1997, and Alexander et al. (2020) reported a global lifetime of inorganic nitrate ($HNO_3$ + total $NO_3^-$) for the period 2000-2015 on the order of 3–4 days. Unfortunately, none of these literatures reported lifetime of coarse nitrate specifically.

The difference in lifetimes of fine and coarse nitrates may be explained by their different formation pathways. Fine nitrate is only formed when there is free $NH_3$ available, and its distribution is concentrated on densely-populated areas (e.g., East Asia, Europe, and eastern America) due to the short lifetime of $NH_3$. The localized production, unstable chemical nature, and efficient deposition of fine nitrate mean that it accumulates only over a relatively short period and on a small spatial scale. In contrast, the formation of coarse nitrate requires availability of $HNO_3$ and coarse particles. The former can be produced from urban emissions of $NO_x$, longer-lived reservoir species of OXN, lightning, and shipping emissions of $NO_x$. The latter is available over both oceans (e.g., sea-salt particles) and continents (e.g., dust particles). As a result, coarse nitrate formation is not limited to a small area but is formed in most areas globally. The chemical stability of coarse nitrate also contributes to its accumulation in the atmosphere, despite its deposition. Existing Figure S1 in the Supplementary also demonstrates a larger abundance of coarse nitrate globally except for a few urban regions, which is again consistent with the longer transport distance (because of longer lifetime) of coarse nitrate compared with fine nitrate.

To provide readers with more information on this topic, we have added the following text and corresponding references in our revised paper (lines 541-547):

"The dominance of coarse nitrate in global total nitrate burden is also observed in other global modelling studies. Hauglustaine et al. (2014), Xu and Penner (2012), and Bauer et al. (2007) reported a relative contribution of coarse-mode nitrate of 72%, 77%, and 79% to total nitrate burden in their respective models for 2000. Additionally, the lifetime of fine nitrate calculated in our work is comparable with other literature values. Park et al. (2004) reported a global fine nitrate lifetime against deposition of 3.2 days for 2001, Feng et al. (2007) reported a global lifetime of total nitrate (fine + coarse) of 5.0 days

for 1997, and Alexander et al. (2020) reported a global lifetime of inorganic nitrate ($HNO_3$ + total $NO_3^-$) for the period 2000-2015 on the order of 3–4 days."

L563-566: Since Ge et al. (2021) showed that the ECLIPSE 2010 $NH_3$ emissions in east Asia were significantly higher than the HTAP 2010 emissions, some of the RDN flux difference compared to Tan et al. (2018) is due to inconsistent emission budgets rather than an actual increase in emissions.

Response: We thank the reviewer for making this useful suggestion, with which we agree. We have double-checked our model results using both HTAP and ECLIPSE emission inventories. The regional budgets of RDN and OXN in East Asia are listed in Table R2. Our RDN and OXN fluxes from HTAP are in similar ranges to values reported by Tan et al. (2018) (RDN: 7.7-8.5 TgN yr$^{-1}$; OXN: 10.5-11.4 TgN yr$^{-1}$) considering that the definitions of East Asia in the two studies do not exactly correspond. In our model experiments, results using ECLIPSE 2010 show higher emissions and deposition for RDN compared to using HTAP 2010 in East Asia, whilst for OXN the differences are only small. The comparison when using the ECLIPSE inventories for 2010 and 2015 shows an increasing trend for RDN fluxes and a decreasing trend for OXN fluxes from 2010 to 2015, albeit the difference between the two ECLIPSE inventories is not as large as that between Tan et al. (2018) and our work.

We have added the following statement to our revised manuscript (lines 605-608):

"Such difference in RDN and OXN fluxes is partially attributed to differences in emission inventories, model structures, meteorology input, and region definition between the two studies. It may also reflect an increase in $NH_3$ emissions and a reduction in $NO_x$ emissions in this region between 2015 and 2010."

**Table R2. RDN and OXN emissions and deposition fluxes in East Asia from our model simulations using HTAP emission inventory in 2010, and ECLIPSE emission inventories in 2010 and 2015.**

| Unit: TgN | HTAP 2010 | ECLIPSE 2010 | ECLIPSE 2015 |
|---|---|---|---|
| RDN emission | 7.50 | 12.0 | 12.5 |
| RDN deposition | 7.32 | 11.4 | 12.0 |
| OXN emission | 10.3 | 10.5 | 9.59 |
| OXN deposition | 8.94 | 9.02 | 8.33 |

Table S5: (1) Given the detailed discussion of this table in the main text, consider moving the table to the main paper as well.

Response: Requested change made.

Table S5: (2) The lifetimes of RDN and OXN in Rest of World as calculated from sources are highly skewed by inputs from other regions. I would suggest removing these numbers and making a note to that effect.

310

Response: We have copied Table S5 to the main paper as a new Table 3. The source-derived lifetimes for Rest of World are removed from Table 3 and a note is added to the caption explaining the reason for that. Table S5 is retained in the Supplement as a record of full data.

315  L656: I suggest adding "and deposition" after "RDN concentrations" given the impact of excess N is a driver for this reduction.

Response: Requested change made.

320  L714-715: While models do have these advantages, it's also important to note their limitations and uncertainties.

Response: We thank the reviewer for making this useful suggestion. In response, we have added the following sentence to the end of the Conclusions of our revised manuscript (lines 778-780):

325

"The chemical and meteorological linkages between emissions and $N_r$ and $S_r$ species concentrations are complex, and the detail can only be revealed by process-based atmospheric modelling as illustrated here, despite the uncertainties in emissions and model parameterisations."

330  **Technical corrections:**

L578: Typo in "deposition".
Response: Requested change made.

335  L650: Remove "The".
Response: Requested change made.

340

**References**

Alexander, B., Sherwen, T., Holmes, C. D., Fisher, J. A., Chen, Q., Evans, M. J., and Kasibhatla, P.: Global inorganic nitrate production mechanisms: comparison of a global model with nitrate isotope observations, Atmos. Chem. Phys., 20, 3859-3877, 10.5194/acp-20-3859-2020, 2020.

Bauer, S. E., Koch, D., Unger, N., Metzger, S. M., Shindell, D. T., and Streets, D. G.: Nitrate aerosols today and in 2030: a global simulation including aerosols and tropospheric ozone, Atmos. Chem. Phys., 7, 5043-5059, 10.5194/acp-7-5043-2007, 2007.

Bellouin, N., Rae, J., Jones, A., Johnson, C., Haywood, J., and Boucher, O.: Aerosol forcing in the Climate Model Intercomparison Project (CMIP5) simulations by HadGEM2-ES and the role of ammonium nitrate, Journal of Geophysical Research: Atmospheres, 116, https://doi.org/10.1029/2011JD016074, 2011.

Feng, Y., and Penner, J. E.: Global modeling of nitrate and ammonium: Interaction of aerosols and tropospheric chemistry, Journal of Geophysical Research: Atmospheres, 112, https://doi.org/10.1029/2005JD006404, 2007.

Ge, Y., Heal, M. R., Stevenson, D. S., Wind, P., and Vieno, M.: Evaluation of global EMEP MSC-W (rv4.34) WRF (v3.9.1.1) model surface concentrations and wet deposition of reactive N and S with measurements, Geosci. Model Dev., 14, 7021-7046, 10.5194/gmd-14-7021-2021, 2021.

Hanke, M., Umann, B., Uecker, J., Arnold, F., and Bunz, H.: Atmospheric measurements of gas-phase HNO3 and SO2 using chemical ionization mass spectrometry during the MINATROC field campaign 2000 on Monte Cimone, Atmos. Chem. Phys., 3, 417–436, https://doi.org/10.5194/acp-3-417-2003, 2003.

Hauglustaine, D. A., Balkanski, Y., and Schulz, M.: A global model simulation of present and future nitrate aerosols and their direct radiative forcing of climate, Atmospheric Chemistry and Physics, 14, 11031-11063, 10.5194/acp-14-11031-2014, 2014.

Park, R. J., Jacob, D. J., Field, B. D., Yantosca, R. M., and Chin, M.: Natural and transboundary pollution influences on sulfate-nitrate-ammonium aerosols in the United States: Implications for policy, Journal of Geophysical Research: Atmospheres, 109, https://doi.org/10.1029/2003JD004473, 2004.

Pringle, K. J., Tost, H., Message, S., Steil, B., Giannadaki, D., Nenes, A., Fountoukis, C., Stier, P., Vignati, E., and Lelieveld, J.: Description and evaluation of GMXe: a new aerosol submodel for global simulations (v1), Geosci. Model Dev., 3, 391-412, 10.5194/gmd-3-391-2010, 2010.

Tan, J., Fu, J. S., Dentener, F., Sun, J., Emmons, L., Tilmes, S., Sudo, K., Flemming, J., Jonson, J. E., Gravel, S., Bian, H., Davila, Y., Henze, D. K., Lund, M. T., Kucsera, T., Takemura, T., and Keating, T.: Multi-model study of HTAP II on sulfur and nitrogen deposition, Atmos. Chem. Phys., 18, 6847-6866, 10.5194/acp-18-6847-2018, 2018.

Xu, L., and Penner, J. E.: Global simulations of nitrate and ammonium aerosols and their radiative effects, Atmos. Chem. Phys., 12, 9479-9504, 10.5194/acp-12-9479-2012, 2012.

385

---

## Author Comment (AC2)

**acp-2022-82: A new assessment of global and regional budgets, fluxes and lifetimes of atmospheric reactive N and S gases and aerosols**

Ge et al.

**Response to Reviewer #2**

We thank the reviewer for their time spent reading our manuscript. Below we include all the reviewer comments and provide in blue text our point-by-point responses. All line numbers mentioned in our responses refer to the clean revised manuscript (not the track-changed version).

**General comments:**

The manuscript provides a comprehensive analysis of the concentrations, wet and dry deposition, fluxes, and lifetimes of reactive N and S gases and aerosols globally, and for 10 world regions. The goal of the manuscript is to conduct a more up-to-date examination of atmospheric processes affecting the fate of reactive N and S, particularly of the reduced N species $NH_3$ and $NH_4^+$, for developing mitigation strategies according to different regional conditions. Overall, this study is an important contribution to the field as it highlights the substantial regional variation in Nr and Sr budgets and the need for modelling to simulate the chemical and meteorological linkages underpinning atmospheric responses to precursor emissions. The manuscript is well written and structured clearly and fits well the scope of this journal. I recommend that this study can be accepted for publication after the specific comments below are addressed.

Response: We thank the reviewer for their supportive comments for our work and for their recommendation of publication after attention to some minor revisions.

**Major comments:**

1. Section 3.1.2, lines 225-228: What was the rationale for choosing the fine SIA concentration of 0.5 µg m⁻³ as the threshold?

Response: The intention of applying such a threshold is to help readers concentrate on regions with relatively larger values of fine SIA concentration where the knowledge of chemical domain is important for formulating anthropogenic emission reduction policies to mitigate the concentrations of $NH_4^+$, $SO_4^{2-}$ and fine $NO_3^-$. As no such threshold value is reported in other studies, we have made our own arbitrary choice based on visualisation of the global SIA distribution. We also trialled other threshold values

(e.g., 0.1 µg m$^{-3}$, 0.7 µg m$^{-3}$, 1.0 µg m$^{-3}$, etc.) and chose 0.5 µg m$^{-3}$ as an appropriate option to present Fig. 3. Perhaps our choice can serve as a reference for future studies.

40   2. Section 3.3.1, lines 460-464: To increase study reproducibility and transparency, the authors may want to consider providing detailed calculation equations of atmospheric burdens and lifetimes (perhaps adding a section in the supplementary).

Response: We thank the reviewer for making this useful suggestion. We have added a new section in
45   the Supplement (Sect. S1.2) that describes the detail of the calculation procedures for atmospheric burdens and lifetimes used in this manuscript. Reviewer #1 also requested more information on this point so see also our responses to this reviewer for more information.

3. Section 3.3.1, lines 517-520: In general, gaseous NH$_3$ deposits quicker than aerosol-phase NH$_4^+$.
50   Could you provide more information on the difference between NH$_3$ deposition and NH$_4^+$ deposition in the model as deposition is crucial for lifetime calculations?

Response: Both NH$_3$ and NH$_4^+$ have dry and wet deposition. The dry deposition of gaseous NH$_3$ to the surface is modelled using the dry deposition velocity parameter. In the model, the deposition velocity
55   is calculated at the centre of the lowest grid (~25 m). Different landcover types (e.g., crops, forests, grassland, seminatural, etc.) have different deposition velocities. For grids containing multiple landcover types, the grid-average deposition rate is calculated based on fractional coverage of different land-cover types within each surface grid.

60   The NH$_4^+$ dry deposition follows the basic formulation for fine-mode aerosol components in EMEP model. Its deposition velocity is dependent on landcover categories and the atmosphere stability (Obukhov length) of turbulent flows in the lower boundary layer. For example, NH$_4^+$ in forests with different Leaf Area Index has different deposition velocity. For stable and unstable stratification conditions, its surface deposition velocity is also different. Detailed formulation is described in Simpson
65   et al. (2012) as cited in the main paper.

Parameterisation of the wet deposition processes in the EMEP model includes both in-cloud and sub-cloud scavenging of gases and particles. As both NH$_3$ and NH$_4^+$ are soluble species, their in-cloud scavenging, following the same equation, is correlated with their own atmospheric mixing ratio and in-
70   cloud scavenging ratio. The parameters of precipitation rate, characteristic scavenging depth, and water density are the same for the two species. For below-cloud scavenging, gaseous NH$_3$ and particle-phase NH$_4^+$ follows two different equations. The former is controlled by its mixing ratio and sub-cloud scavenging ratio, while the latter is controlled by its mixing ratio, the raindrop fall speed, and the sizedependent collection efficiency of aerosols by the raindrops. Again, the detailed formulation is described in the Simpson et al. (2012) citation in the paper.

4. Section 3.3.3, lines 618-619: Is there any explanation for why does South America have such large proportion of RDN in the form of $NH_3$?

Response: This may be related to South America's distinctive $NH_3$ emission characteristic. Chen et al. (2013), Whitburn et al., (2014), van Marle et al. (2017) and Evangeliou et al. (2021) all showed that natural $NH_3$ emissions from forest, savanna, and agriculture fires are more significant in this region compared to other world regions. $NH_3$ emitted from biomass burnings is mainly concentrated on remote or rural areas, and away from anthropogenic emissions of $NO_x$ and $SO_x$. As a result, most $NH_3$ stays in the gaseous phase and quickly deposits back to ground without forming stable $NH_4^+$ aerosol (i.e., $(NH_4)_2SO_4$; $NH_4NO_3$ is easily converted back to $NH_3$ under high temperature conditions), which explains the large proportion of $NH_3$ in total RDN burden over South America. Our discussion of regional RDN deposition in Sect. 3.2.1 also shows that RDN deposition in South America is dominated by $NH_3$ (86%; Fig. 5), which is consistent with our explanation here.

**Technical corrections:**

Line 758: DOI information is missing in this citation. Please add the "doi" number before "2013b".
Response: Requested change made.

Line 809: Please add the "doi" number before "2005".
Response: Requested change made.

Line 896: Please add the "doi" number before "2015".
Response: Requested change made.

Line 966: Please add the "doi" number before "2012".
Response: Requested change made.

**References**

Chen, Y., Morton, D. C., Jin, Y., Gollatz, G. J., Kasibhatla, P. S., Van Der Werf, G. R., Defries, R. S., and Randerson, J. T.: Long-term trends and interannual variability of forest, savanna and agricultural fires in South America, Carbon Manag., 4, 617–638, https://doi.org/10.4155/cmt.13.61, 2013.

Evangeliou, N., Balkanski, Y., Eckhardt, S., Cozic, A., Van Damme, M., Coheur, P. F., Clarisse, L., Shephard, M. W., Cady-Pereira, K. E., and Hauglustaine, D.: 10-year satellite-constrained fluxes of ammonia improve performance of chemistry transport models, Atmos. Chem. Phys., 21, 4431-4451, 10.5194/acp-21-4431-2021, 2021.

van Marle, M. J. E., Field, R. D., van der Werf, G. R., Estrada de Wagt, I. A., Houghton, R. A., Rizzo, L. V., Artaxo, P., and Tsigaridis, K.: Fire and deforestation dynamics in Amazonia (1973–2014), Global Biogeochem. Cy., 31, 24–38, https://doi.org/10.1002/2016GB005445, 2017.

Whitburn, S., Van Damme, M., Kaiser, J. W., Van Der Werf, G. R., Turquety, S., Hurtmans, D., Clarisse, L., Clerbaux, C., and Coheur, P. F.: Ammonia emissions in tropical biomass burning regions: Comparison between satellite-derived emissions and bottom-up fire inventories, Atmos. Environ., 121, 42–54, https://doi.org/10.1016/j.atmosenv.2015.03.015, 2014.